# Adjustments to an abrupt solar forcing in the CMIP6 abrupt-solm4p experiment

Charlotte Lange [1] and Johannes Quaas [1]

[1]Leipzig Institute for Meteorology, Leipzig University, Stephanstraße 3, 04103 Leipzig, Germany

**Correspondence:** Charlotte Lange  (charlotte.lange@uni-leipzig.de)

**Abstract.** The concept of "radiative" or "rapid" adjustments refers to the response of the climate system to an instantaneous radiative forcing, independent of surface temperature changes. These adjustments can occur over time scales from hours (e.g. aerosol-cloud-interactions) to months (e.g. stratospheric temperature changes) or even longer, making it challenging to distinguish adjustments from feedbacks over longer time scales. Despite variations in definitions, understanding these processes is essential for advancing climate modelling.

Adjustments to radiative forcing are an important yet uncertain component of climate forcings. However, disentangling adjustment and feedback mechanisms in realistic scenarios remains highly challenging. Therefore, analysis of more controlled experiments can shed light onto different kinds of adjustment mechanisms, improve understanding and can be an important contribution on our way to more realistic scenarios like volcanic eruptions or the current climate change.

The abrupt-solm4p experiment within the Cloud Feedback Model Intercomparison Project (CFMIP) as part of the 6th Coupled Model Intercomparison Project (CMIP6) simulates an instantaneous $4\%$ reduction in the solar constant, starting from a pre-industrial run on 1 January 1850. This study analysed changes in climate variables, cloud properties, and radiative fluxes over different time scales (hours, days, months and up to 150 years) to understand adjustment processes.

Four models were evaluated, showing initial rapid cooling, particularly over Antarctica and the southern hemisphere, slowing down the polar night jet, disrupting the polar vortex and increasing Arctic cloud cover. During the first month, the troposphere cools down faster than the ocean surface, decreasing vertical stability and increasing cloud cover over ocean, while the opposite effect happens in the tropics over land. This in turn affects land-sea-circulation. On longer time scales we find robust changes of cloudiness.

# 1  Introduction

The state of the Earth climate is determined by the radiative fluxes entering and leaving the atmosphere. In a steady-state system incoming and outgoing radiative energy fluxes are equal and the energy budget of the Earth climate system is in balance. In case of any perturbation of the energy budget (radiative forcing), the climate system reacts by heating or cooling, which leads to a new balance of energy fluxes on time scales of centuries. Different processes in the climate system either enhance or dampen the Earth's capability to reach a new equilibrium. Such processes act either in response to global mean surface temperature change, and typically on longer time scales, the so-called feedbacks, or they are independent of surface temperature change and mostly act on shorter time scales of days, weeks and months, the so-called adjustments. Radiative adjustments of the atmosphere to an external forcing are of particular interest to the scientific community and a growing number of studies on the subject is conducted (Gregory et al., 2004; Zelinka et al., 2013; Myhre et al., 2013; Sherwood et al., 2015; Smith et al., 2018; Forster et al., 2021; Quaas et al., 2024). By introducing radiative adjustments into the forcing-feedback-framework as part of the effective radiative forcing (ERF, Myhre et al., 2013), the resulting ERF is a better predictor of implied global surface temperature change, i.e. more independent of the kind of forcing agent (Andrews and Forster, 2008; Gregory and Webb, 2008). However, this approach requires a precise estimate of ERF, which comprises the instantaneous radiative forcing (IRF) as well as (radiative or rapid) adjustments (RA), which are prone to high uncertainty, especially in case of cloud adjustments (Andrews and Forster, 2008).

The total global mean radiative imbalance at the top of atmosphere (TOA) $\bar{N}(t)$ can be written as

$$\bar{N}(t) = IRF + RA + \sum_i \lambda_i \Delta \bar{T}(t) \tag{1}$$

where the sum over the feedback parameters $\lambda_i$ accounts for the different sources of feedbacks.

Short-term RA comprise all adaptations of the atmosphere to a radiative forcing, that are independent of a global mean surface temperature change (Andrews and Forster, 2008; Gregory and Webb, 2008; Sherwood et al., 2015). However, local changes of surface temperature and consequential atmospheric adjustments can contribute a considerable amount to the overall RA (Quaas et al., 2024). In many cases, these adjustments happen on time scales from hours to days, in case of precipitation and cloud adjustments, but adjustments of the stratosphere, cryosphere or vegetation can take months to years (Forster et al., 2021; Stjern et al., 2023). This leads to an overlap of adjustment and feedback time scales, which makes it difficult to disentangle both. Moreover, adjustments are often confounded by climate variability in magnitude, making it very hard to detect them in observations.

This is further complicated by the transient nature of most forcing processes happening in the Earth climate system, like the gradual increase of $CO_2$ from anthropogenic sources in the atmosphere. In contrast to that, climate models allow the application of instantaneous forcings, that exceed climate variability and are kept constant for longer time scales allowing for a better analysis of adjustments and feedbacks. A common experiment using global climate models is the instantaneous two- or fourfold increase of the atmospheric $CO_2$ concentration (e.g., Gregory et al., 2004; Colman and McAvaney, 2011; Kamae

and Watanabe, 2012; Zelinka et al., 2013; Andrews et al., 2015) in order to predict long-term consequences of anthropogenic climate change as well as short-term adjustments. Special focus has been on adjustments of cloud optical properties and cloud fraction, since they are an important source of uncertainty in climate models. Reducing short-term uncertainty of cloud adjustments could also reduce uncertainty of long-term predictions (Andrews and Forster, 2008; Nam et al., 2018). Several studies have found a positive shortwave adjustment in 2x or 4xCO$_2$-simulations due to the reduction of cloud fraction in tropical and mid-latitudes associated with decreased relative humidity in reaction to the warming of the lower troposphere (e.g., Gregory and Webb, 2008; Colman and McAvaney, 2011).

However, these simulations are often highly idealised, using fixed sea-surface temperatures or omitting the existence of continents (aqua-planet-simulations). In order to compare and validate such results with observations, a natural forcing is required, that is applied nearly instantaneously and is of sufficient strength. Volcanic eruptions are considered such so-called "natural laboratories" (e.g., Malavelle et al., 2017; Christensen et al., 2022). During large volcanic eruptions, huge amounts of aerosols are emitted into the atmosphere in a very short time frame of hours. If transported up to the stratosphere, depending on the location and season of the volcanic eruption, the aerosol can form a scattering layer around the globe with a lifetime of up to one to three years (Myhre et al., 2013). This way, volcanic eruptions exert a radiative forcing, that is approximately comparable to the instantaneous forcing that usually only models can realise. However, examining volcanic eruptions includes other challenges. The initial forcing is very localised and only after a few months the aforementioned scattering layer is formed around the globe. Depending on the location of the volcano, the strength of the manifold adjustment mechanisms and the distribution of the stratospheric aerosol layer varies. Hence, time scales of different adjustments might still overlap. Therefore, before analysing adjustments to a realistic volcanic eruption, it can be helpful to further simplify the problem.

In this study, we thus examine the results of the abrupt-solm4p experiment, which is part of the Cloud Feedback Model Intercomparison Project (CFMIP) as part of the 6th Climate Model Intercomparison Project (CMIP6) (Eyring et al., 2016; Webb et al., 2017). The output of four global climate models, which participated in this experiment, is available. The simulation reduced the solar constant instantly by $4\%$. This can be regarded as a simplified analogue for an aerosol scattering layer. Both, volcanic eruptions and a reduced solar constant lead to a reduced incoming shortwave flux at the surface, even if stratospheric adjustments are expected to differ. Hence, similar adjustment patterns are expected for both types of forcing, when the scattering sulfate layer distributes over the whole globe with time. Several studies analysed the reaction of the Earth climate system to a number of different forcing agents (e.g., Gregory et al., 2004; Smith et al., 2018), however only few studies quantified radiative adjustments after solar forcing and took a closer look at the specific processes happening in response to solar forcing (e.g., Salvi et al., 2021; Virgin and Fletcher, 2022; Aerenson et al., 2024).

Radiative adjustments to solar forcing in sum typically counteract the initial forcing (e.g., Smith et al., 2018; Russotto and Ackerman, 2018; Virgin and Fletcher, 2022; Aerenson et al., 2024). Most of this is attributed to decreased temperature of troposphere, stratosphere and land surface, which leads to a reduction in longwave radiation lost to space. The effect is partially offset by the reduction of water vapour in the atmosphere, following the Clausius-Clapeyron relationship, and thereby an increase in outgoing longwave radiation. Moreover, Smith et al. (2018) derive a positive adjustment due to change in surface albedo.

However, when analysing rapid adjustments due to changes in cloud properties, the studies find contradictory results. Smith et al. (2018) and Virgin and Fletcher (2022) find cloud adjustments that counteract the initial solar forcing. In response to a $+2\%$ solar constant forcing, Smith et al. (2018) describe an increase in cloud fraction in the boundary layer, a reduction of cloud fraction in the free troposphere, a small increase in cloud fraction at around $100\,\mathrm{hPa}$ and a decrease above this level. This is very similar to our findings, although of the opposite sign, since we examined the response to a reduction of solar constant. Using radiative kernels, Smith et al. (2018) show that cloud changes lead to negative long- and shortwave effects, but with high model disagreement in the shortwave effects. In response to a $+2\%$ solar forcing, Virgin and Fletcher (2022) find that the shortwave component is dominated by changes in the boundary layer clouds, while changes in the troposphere contribute more to the longwave effects. In contrast to this, Aerenson et al. (2024) find cloud adjustments that further amplify the initial solar forcing. In response to a $+4\%$ solar constant forcing, they find an overall positive cloud radiative adjustment, which is strongest, where low-cloud fraction is reduced. The two main mechanisms are changes in cloud fraction and cloud optical depth, which they find are of opposite sign in case of solar forcing. Moreover, short- and longwave effects tend to cancel each other out, but shortwave effects are slightly stronger. Hence, in total, positive shortwave effects due to decreased cloud fraction dominate the overall cloud adjustments. However, they find that the participating models do not agree on the signs of cloud adjustment effects and the uncertainty of the total effect is high.

Even though the studies describe similar changes in cloud properties, the derived cloud radiative adjustments differ depending on the applied method and models used. This demonstrates, how radiative effects of cloud property changes are still one of the major sources of uncertainty in climate models, even in highly idealised solar forcing scenarios.

Reduced solar constant simulations are also of interest for the increasing number of solar radiation modification (SRM) studies, that often aim to balance the forcing due to anthropogenic emissions of $CO_2$ by reducing the absorbed solar radiation (e.g., Bala et al., 2008; Cao et al., 2015; Schmidt et al., 2012; Huneeus et al., 2014; Russotto and Ackerman, 2018). Having a better understanding of not only the long-term effects of SRM methods, but also knowing about short-term adjustments, can improve the risk assessment of these endeavours. In these scenarios, not only the global mean development of climate variables, but also, if not even more, local processes are of high importance. Especially short-term adjustments of clouds and precipitation can lead to local droughts or floods, with possibly important effects for local communities. Hence, a better understanding of adjustments to certain types of forcing, will not only reduce the uncertainty of long-term predictions of climate models but also improve short-term forecast.

There are several methods to quantify radiative adjustments in climate models. On the one hand, the linear regression method (Gregory et al., 2004) allows for a simultaneous determination of ERF, equilibrium climate sensitivity (long-term trend) and climate feedback parameter ($\sum \lambda_i$). In order to derive radiative adjustments from ERF, knowledge of the instantaneous forcing is necessary. For most forcing agents, it is possible to diagnose the IRF by separate radiative transfer modelling. However, the regression method is only feasible for the global mean and it relies on the assumption, that all rapid adjustments happen, while the global mean surface temperature change is still zero. This is an oversimplification, as inert systems like cryosphere or vegetation might still adjust to the initial forcing, while global mean surface temperature already begins to change.

On the other hand, the fixed surface temperature method (Hansen et al., 2005) or rather fixed-sea-surface temperature (SST), which is easier to implement in global climate models (Forster et al., 2016), is widely used and has the advantage of suppressing feedbacks. This allows for the disentanglement of adjustments and feedbacks. However, this method artificially suppresses adjustments of ocean surface temperature and introduces unrealistic land sea contrast, which hinders a realistic estimate of circulation adjustments. Andrews et al. (2021) showed a significant difference in ERF depending on whether only sea surface

temperature or all surface temperature was kept to zero. Possible adjustments to localized warming or cooling, cannot be simulated in this kind of setups, although in some concepts they are considered adjustments relevant to TOA ERF (Quaas et al., 2024).

      The simulations analysed in this study apply an instantaneous solar forcing that is kept constant, while allowing the whole climate system to adapt in a fully coupled general circulation model. The regression method was used to quantify ERF and RA,

but non-linear behaviour was found for the first 4-10 years, when the more inert systems like ocean, cryosphere or vegetation still adjust to the forcing, while global mean surface temperature change starts to dampen TOA effective flux imbalance.

      Nevertheless, this model setup allows us to examine adjustment processes on time scales of hours, days and months, where, as we show, TOA effect flux change is still dominated by adjustment processes rather than temperature mediated change.

      In a reduced solar constant experiment a variety of adjustment processes can be analysed. Thermodynamical and dynamical

adjustments due to cooling or heating of the atmosphere, but also, visible in the anomaly of surface and atmospheric temperature, lead to changes in pressure systems and circulation patterns. Changes in cloud properties and precipitation during the first hours are a result. On longer time scales also a change in vegetation might happen, but is not subject of this study and no further analysis was performed. An important focus of this study are cloud adjustments, which even in this simple experiment are clearly the source of highest uncertainty. The aim of this study is to identify typical response patterns as well as global

mean radiative adjustments to solar forcing.

      Section 2 contains a more detailed description of the experiment setup as well as an overview of the available data. Section 3 first shows the results of the classical regression method by Gregory et al. (2004), in order to quantify ERF and RA. It then discusses, how time scales of hours, days and months are dominated by adjustment processes, to proceed with a closer analysis of adjustment processes of a number of different climate variables. These include surface and atmospheric temperature,

relative humidity, vertical velocity and cloud fraction. In the end, we take a closer look on the radiative flux anomalies and finally analyse cloud adjustments by examining adjustments in the cloud radiative effect. Section 4 discusses the results of this study in the context of studies that have been conducted on adjustments to solar forcing in recent years and discusses the limitations of the dataset and the approach. Section 5 contains a summary and provides an outlook on how this studies findings can contribute to future research in the field of radiative adjustments.

## 2   Methods

In the solm4p-experiment, the solar constant is reduced instantaneously by $4\,\%$ and kept constant at $96\,\%$ of the solar constant of the control run (pi-Control). It branches off the pi-Control simulation on 1 January 1850. No other forcing is considered in

the experiment. The diagnostics of four models, which participated in this experiment, are available: IPSL-CM6A-LR, CESM2, CanESM5 and MRI-ESM2-0 (further information provided in Table 1). For each model one experiment run corresponding to

155 one pre-industrial control run (piControl) is provided. While some models provide 3-hourly output for several parameters, CESM2 only provides daily data. In case of vertically resolved atmospheric cloud fraction, only monthly data was available from all four models. More information on the four models, including their horizontal and vertical grid spacing is provided in Table 1.

For all variables examined in this study, the difference attributable to the reduction in solar constant was calculated for each

160 parameter by subtracting the piControl-run from the solm4p-run for each point in time and space.

Following Stjern et al. (2023), four different time scales were considered: the first 100 hours, the first 30 days, the first year and the following years until 150 years after the onset of forcing. For the first time scale (up to 100 hours) 3-hourly data was used, when available, else daily data was used. For the second time scale (days 5–30) daily averages were used, for the third time scale (months 2–12) monthly data was used and for the long-term development (years 2–150) yearly means were plotted.

The transitions between the plots of different time scales often display a small jump, because the next frequency does not contain the last datapoint of the preceding frequency, but an average.

For all time scales the global mean was calculated for each model. Moreover, a multi-model mean was calculated by first interpolating all models to the time axis of the MRI-ESM2-0 model. This model was chosen as reference, because it provided the most extensive database for the abrupt-solm4p experiment. If not all models provided data of the same frequency, the

170 highest available resolution was plotted for each model, respectively and for the multi-model mean, the other models were interpolated to the 3-hourly time axis of MRI-ESM2-0.

In addition to the temporal development of global means, a geographical distribution of the respective parameter was plotted, averaging over the respective time scales. For the mean of the first three time scales, all time steps were averaged, while for the fourth time scale only the years 120-150 after the onset of forcing were averaged to obtain an estimate for the long-term

new approximate equilibrium state. For the global distribution, a multi-model mean was calculated by interpolating the other models to the CanESM5 horizontal grid, which was the coarsest out of the four models.

Besides the grid spacing, also the height levels varied between the models in case of the atmospheric cloud fraction data. For the other climate variables, all models used the same 19 height levels. In case of cloud fraction, the other models were interpolated to the CanESM5 pressure axis.

In order to account for uncertainty, the multi-model standard deviation was plotted together with the temporal development of multi-model global mean. In case of global distributions of anomalies, areas were dotted in which less than three of four models agreed on the sign of anomaly.

To analyse differences in land and sea surface response, some climate variables, e.g. relative humidity, were averaged only over ocean and only over land. This was done by applying a land-sea-mask, based on the CanESM5 grid, before performing

the zonal averaging.

This study focuses on cloud adjustments and the cloud radiative effect, which was calculated from the difference between simulation runs with clouds (all-sky) and a second radiative transfer calculation for each step without clouds (clear-sky), both

| | CanESM5 | CESM2 | IPSL-CM6A-LR | MRI-ESM2-0 |
|---|---|---|---|---|
| **Full name** | Canadian Earth System Model version 5 | Community Earth System Model 2 | Institut Pierre-Simon Laplace - Climate Model 6A -Low resolution | Meteorological Research Institute Earth System Model Version 2.0 |
| **Reference** | Cole et al. (2019) | Danabasoglu (2020) | Boucher et al. (2018) | Yukimoto et al. (2020) |
| **Grid size (lon x lat)** | 128 x 64 | 288 x 192 | 144 x 143 | 320 x 160 |
| **Pressure levels (Pa)** | 8 (daily) 19 (monthly) | 8 (daily) 19 (monthly) | 8 (daily) 19 (monthly) | 8 (daily) 19 (monthly) |
| **Cloud height levels** | 49 (atmosphere hybrid sigma pressure coordinate (unitless)) | 32 (atmosphere hybrid sigma pressure coordinate (unitless)) | 79 (pressure levels (Pa)) | 80 (atmosphere hybrid sigma pressure coordinate (unitless)) |

**Table 1.** Information on the four models for which output was available from the CFMIP abrupt-solm4p-experiment.

provided by each of the four models. Then, the total cloud radiative effect anomaly at TOA was calculated as the combined effect of solar (shortwave) and terrestrial (longwave) radiation changes due to cloud changes.

The linear regression method introduced by Gregory et al. (2004) was applied for a number of TOA fluxes, in order to estimate their influence on the overall ERF. For that purpose, the yearly global mean of the respective TOA flux change was plotted over the yearly global mean of near-surface temperature change and a linear regression was used, which determined the intercept with the TOA flux axis.

## 3 Results

In order to estimate the rapid adjustments to a $4\%$ solar constant reduction, we applied the regression method developed by Gregory et al. (2004) to the solm4p-CFMIP data. The results are shown in the following section. We then move on to examine different time scales after the onset of forcing and motivate, how the first three time scales (hours, days and months) are dominated by rapid adjustments, while the fourth time scale shows the long-term adaptation and is therefore dominated by surface temperature mediated processes. Afterwards, we analyse the response of different climate variables like temperature, humidity and cloud properties on the different time scales and finally examine how these climate variables influence top of atmosphere fluxes and the cloud radiative effect.

### 3.1 Effective radiative forcing and rapid adjustment estimate

We applied a linear regression to the yearly global means of TOA radiative budget change ($\Delta$ TOA budget) and near surface temperature change ($\Delta T$) of the four available models.

The results are shown in Fig. 1a together with the same plots for the individual TOA budget components.

The TOA radiative budget anomaly shows the expected behaviour for all four models, starting with negative values for $\Delta T = 0$ and then approaching a new radiative balance. The development is mostly linear, however, especially for the first 10 years all models exhibit a slightly steeper slope than in long term. This deviation could either be interpreted as radiative adjustments developing as long as a decade or it might suggest, that the overall relation between $\Delta T$ and TOA radiative budget anomaly changes after about a decade, due to the inertia of the ocean. The multi-model mean IRF was estimated as the difference between downward and upward shortwave anomaly for the first month after onset of forcing (shortest output frequency available for all models). The intercept of the linear regressions with the y-axis provides the ERF and the difference between ERF and IRF yields the RA. Multi-model mean ERF and RA are given at the bottom of the figure and are also provided in Table 2. The negative IRF of approximately $-10\,\mathrm{W\,m^{-2}}$ is reduced by radiative adjustments of $3.6\,\mathrm{W\,m^{-2}}$, resulting in an ERF of $6.4\,\mathrm{W\,m^{-2}}$. While some models, like CanESM5 and IPSL-CM6A-LR, simulate a strong temperature decrease in response to the radiative forcing, CESM2 and MRI-ESM2-0 reach a new equilibrium at a significantly weaker change in surface temperature.

Analogous to the total TOA radiative budget, also the yearly mean of the components of the TOA budget, i.e. downward shortwave flux change $\Delta \mathrm{sw}^{\downarrow}_{\mathrm{allsky}}$, upward shortwave flux change $\Delta \mathrm{sw}^{\uparrow}_{\mathrm{allsky}}$ and upward longwave flux $\Delta \mathrm{lw}^{\uparrow}_{\mathrm{allsky}}$, were plotted against the surface temperature change and linear regressions were applied in Figures 1b, 1c and 1d. Again offsets after one month of simulation were marked as red crosses to provide insight into the contribution to overall RA of the single radiative components. The multi-model mean intercepts of the linear regression with the radiative flux axis are shown in the respective figures as well as in Table 2. As determined by the experiment conditions, $\Delta sw^{\downarrow}$ (Fig. 1b) is held constant over the whole experiment time. However, a seasonal variation occurs due the elliptical orbit of the Earth and produces a small offset between the first month value (January) and the yearly mean values. This offset is considered as part of the IRF not the RA.

In contrast, $\Delta sw^{\uparrow}$ (Fig. 1c) exhibits a clear approximately linear relation to $\Delta T$. All models simulate an instantaneous response of $\Delta sw^{\uparrow}$ in reaction to the reduction of solar constant (red cross), depending on the planetary albedo. However, the contribution of $\Delta sw^{\uparrow}$ to the ERF is more negative than this initial value ($-6.3\,\mathrm{W\,m^{-2}}$), indicating shortwave radiative adjustments of $-6.3\,\mathrm{W\,m^{-2}} - (-4)\,\mathrm{W\,m^{-2}} = 2.3\,\mathrm{W\,m^{-2}}$. All models display a deviation from the overall linear behaviour during the first decade. Other studies like Gregory and Webb (2008) found similar non-linearities. However, in their studies those were attributed to shortwave cloud radiative effects, while we found the source of this non-linearity to be in the clear-sky shortwave component over ocean, which is linked to short-term changes in sea-ice extent and snow cover, but were not further analysed in this study. The respective intercepts are provided in Table 2.

Figure 1d shows the same linear regression for upward longwave radiation anomaly ($\Delta lw^{\uparrow}$). All models simulate a clear linear relationship between $\Delta lw^{\uparrow}$ and $\Delta T$ with an intercept of around $-1\,\mathrm{W\,m^{-2}}$, which is the longwave adjustment. Hence, the upward longwave component contributes about $1/3$ to the overall RAs, while the shortwave RAs make up the other $2/3$. In contrast to the upward shortwave component, there is no systematic deviation from linear behaviour in the longwave component during the first decade.

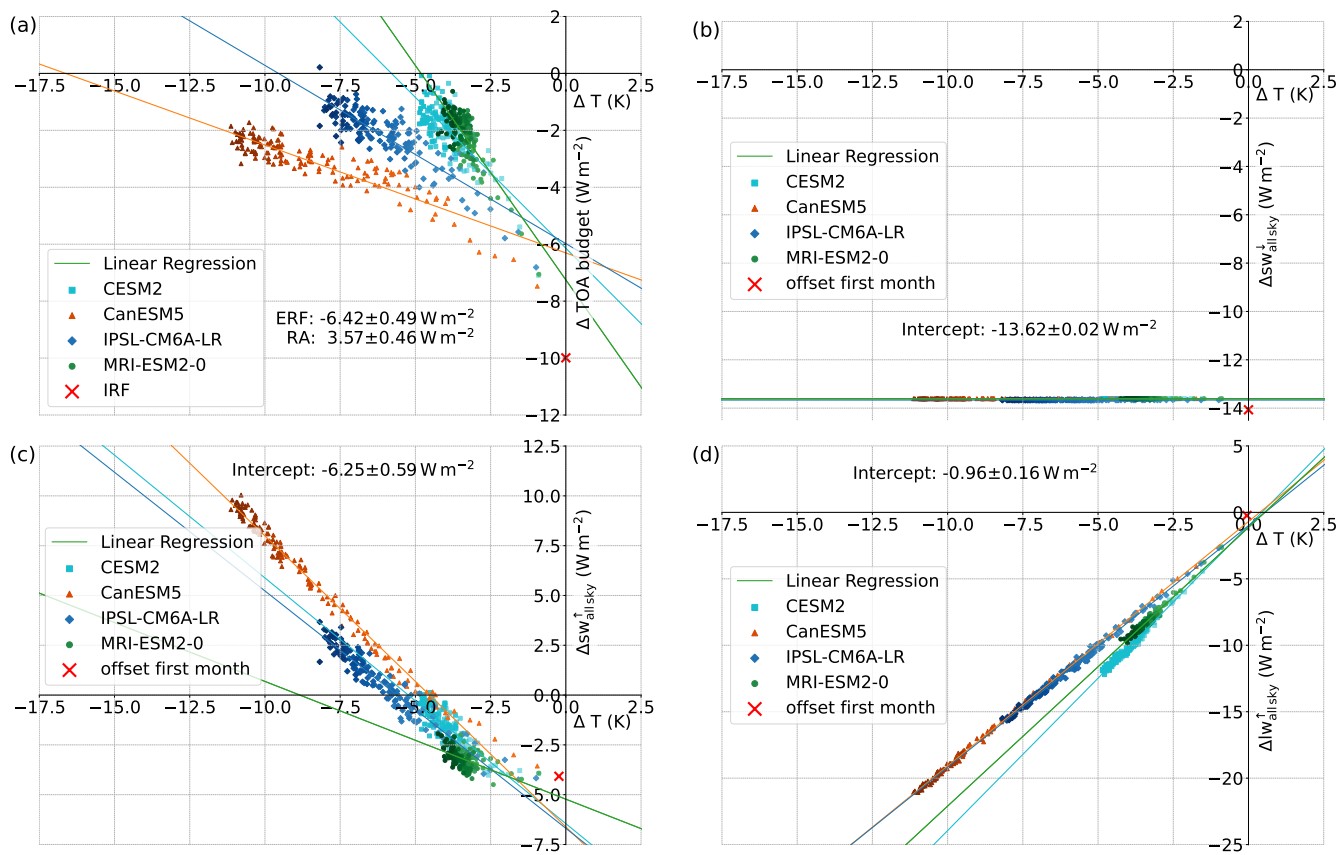

**Figure 1.** Linear regression plots for yearly global mean TOA radiative flux anomalies in $\mathrm{W\,m^{-2}}$ vs. change in yearly global mean surface temperature ($\Delta T$). Plotted are (a) the TOA radiative budget, (b) the downward shortwave flux at TOA, (c) the upward shortwave flux at TOA and (d) the upward longwave flux at TOA. The respective initial offsets after one month are marked as a red cross.

The same method was applied to the cloud radiative effect anomaly ($\Delta\mathrm{CRE_{TOA}}$) and its long- and shortwave components ($\Delta\mathrm{CRE_{TOA,sw}}$ and $\Delta\mathrm{CRE_{TOA,lw}}$), respectively. If an approach, similar to the IRF-ERF approach is applied and the instantaneous CRE is subtracted from the intercept, as it is purely a cloud masking effect, the remaining cloud adjustment is $3.3\,\mathrm{W\,m^{-2}} - 2\,\mathrm{W\,m^{-2}} = 1.3\,\mathrm{W\,m^{-2}}$, mostly stemming from $\Delta\mathrm{CRE_{TOA,sw}}$, which is a substantial contribution to the overall RA of $3.6\,\mathrm{W\,m^{-2}}$. Hence, we find that cloud adjustments counteract parts of the initial solar forcing as was found by several other studies (Smith et al., 2018; Salvi et al., 2021; Virgin and Fletcher, 2022). However, the uncertainty is high and no clear pattern emerged, because models disagreed on the sign of slope of the linear fits. The intercepts for the cloud radiative effect anomaly and its components are also provided in Table 2.

| TOA flux anomalies | ERF component ($\mathrm{W\,m^{-2}}$) |
|:---:|:---:|
| total budget | -6.42 ± 0.49 |
| sw downward (all-sky) | -13.62 ± 0.02 |
| sw upward (all-sky) | -6.25 ± 0.59 |
| lw upward (all-sky) | -0.96 ± 0.16 |
| CRE total | 3.33 ± 0.73 |
| CRE net sw | 3.0 ± 0.59 |
| CRE net lw | 0.33 ± 0.19 |
| sw upward (clear-sky) | -3.42 ± 0.77 |
| lw upward (clear-sky) | -0.63 ± 0.07 |
| sw upward (clear-sky ocean) | -3.16 ± 1.02 |
| sw upward (clear-sky, land) | -0.08 ± 0.28 |

**Table 2.** Multi-model-mean intercepts of different TOA fluxes derived via regression method.

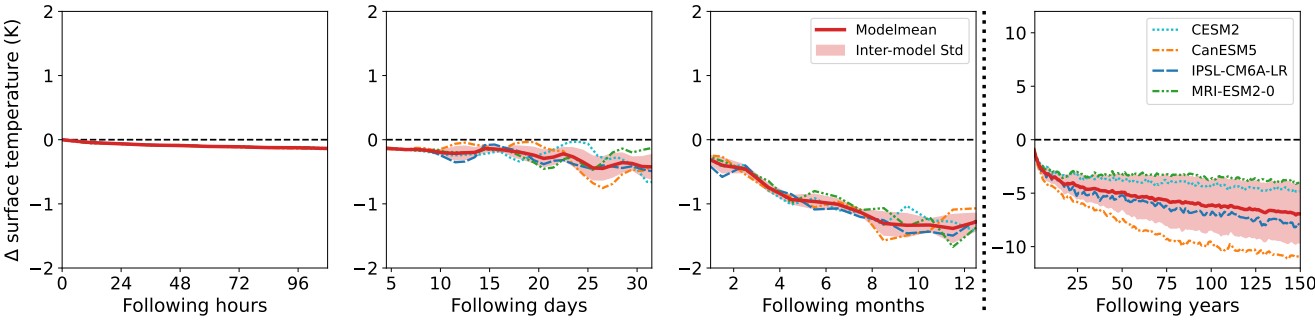

**Figure 2.** Global mean of near surface temperature anomaly in K for four different time scales after the onset of forcing (100 hours, 30 days, 12 month, 150 years). The results of the four participating models are plotted (cyan: CESM2, orange: CanESM5, blue: IPSL-CM6A-LR and green: MRI-ESM2-0) together with a multi-model mean (red, bold) and multi-model standard deviation (shaded area around multi-model mean). For the fourth timescale, the y-axis has been adjusted to account for the stronger changes on longer time scales.

## 3.2 Rapid adjustments on different time scales

In this study, we are interested in rapid adjustments to solar forcing in fully coupled climate models. Hence, the surface temperature begins to adapt from the onset of forcing, thereby possibly overlying adjustments of the atmosphere to the forcing with temperature mediated effects.

The global mean near surface temperature anomaly is shown in Fig. 2. It starts to become increasingly negative immediately after the onset of forcing due to decreased absorption of shortwave radiation. The global mean near surface temperature decrease is of the order of $0.25\,\mathrm{K}$ over the the first month, continuously growing to values $> 1\,\mathrm{K}$ after four months. The

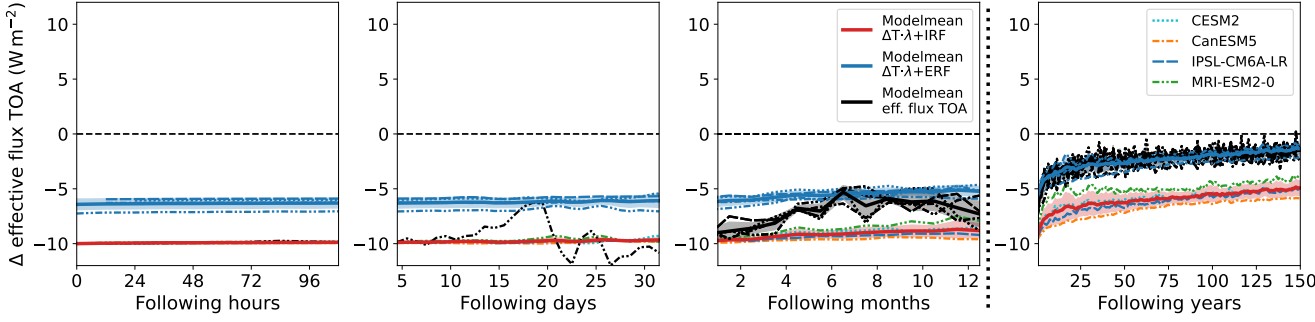

**Figure 3.** Same as Fig. 2, but for TOA effective flux change. Shown are TOA effective flux change, based on the simulated surface temperature change, only considering IRF (red), the calculated TOA effective flux change considering IRF+RA (blue) and the simulated TOA effective flux change (black).

four participating models agree well during the first year in terms of this global average, but differ in their long-term surface
temperature change, depending on the models climate sensitivity.

Changes of the global mean surface temperature of more than $1\,\mathrm{K}$ lead to changes of the TOA radiative forcing. However, when calculating the TOA flux anomaly via Eq. 1 by multiplying the surface temperature change with the feedback parameter and adding the ERF, both derived from the linear regression in Sect. 3.1, a significant disagreement between calculated TOA effective flux anomaly and simulated TOA effective flux anomaly becomes apparent. Figure 3 shows the simulated TOA effec-
260 tive flux anomaly (black) together with the TOA flux anomaly calculated from the simulated global mean surface temperature change and adding only the IRF (red) and the complete ERF (blue) for the four different time scales. During the first time scale no adjustments or feedbacks take place that would change the global mean TOA flux anomaly considerably. Hence, the red and black curves do not show a strong deviation. But on all other time scales there is a significant offset between the two. Especially for the time scales of days and months after the onset of forcing, the trend of surface mediated TOA flux change
cannot explain the trend of the simulated TOA flux anomaly. For the fourth time scale, both model means show the same trend, but with a constant offset. When adjusting the surface temperature mediated TOA flux change in Fig. 3 by the rapid adjustments (blue curve) from Sect. 3.1, the long term trend of calculated and simulated TOA flux change agrees well. However, the first three time scales show considerable deviations between calculated and simulated TOA effective flux change and temperature mediated flux change is clearly not sufficient to explain the simulated TOA flux change. Hence, we argue, that the first three
time scales are dominated by adjustment processes and temperature mediated changes only play a minor part. Clearly, this does not allow for a perfect distinction between adjustments and feedbacks, however, this approach has the advantage that no restrictions have to be made to surface temperature (e.g. fixed SST experiments), which fail to capture all changes in circulation and no assumptions are made on the linearity of adjustment processes, as is done in case of radiative kernels. This study aims to understand changes in the atmospheric state during the first year after onset of forcing and based on the beforehand arguments,
we assume this time scale to be dominated by adjustment processes.

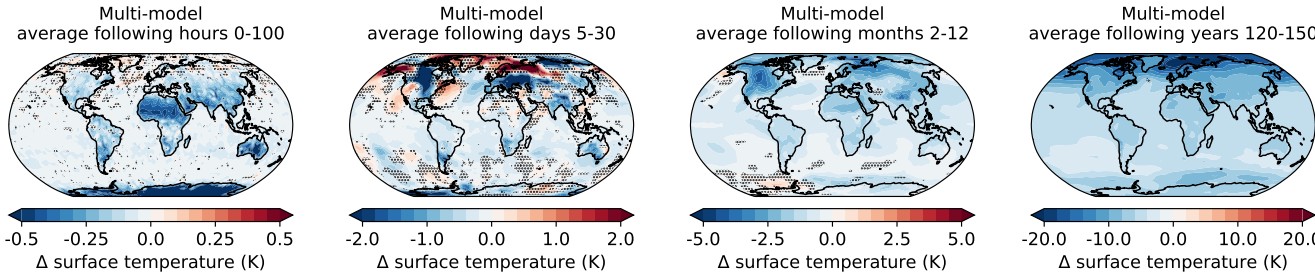

**Figure 4.** Multi-model mean global distribution of near surface temperature anomaly in K averaged over four different time scales after onset of forcing (100 hours, days 5–30, months 2–12, years 120–150). The multi-model mean was calculated by interpolating the other models to the grid spacing of the CanESM5 model. Regions, where less than 3 of 4 models agreed on the sign of anomaly, are dotted. Note the different scales of the colour bars for better contrast.

### 3.2.1 Adjustments of surface and atmospheric temperature

The effects of the reduction of the solar constant are visible in a variety of climate variables already a few hours after the onset of the perturbation.

Figure 4 shows the geographical distribution of the near surface temperature anomaly averaged over the first three time scales depicted in Fig. 2 (hours, days and months, respectively) and a fourth time scale averaged over the years 120–150 representing the long-term development.

During the first 100 hours the land surface responds to the forcing by cooling down, while the ocean surface temperature stays approximately constant due to its higher heat capacity. The strongest cooling occurs over Antarctica, because of the 24 hour exposure to solar radiation at time of forcing onset (1 January). Areas of low heat capacity (e.g. Antarctica and Sahara) generally exhibit a stronger surface temperature decrease compared to other regions on the same latitude, because the same amount of heat reduction will lead to stronger decrease in temperature. In contrast to the overall cooling of surface temperature, the Arctic, which does not experience any solar forcing in January, warms up during the first hours and days after onset of forcing. The reduced temperature gradient between the Tropics and the Arctic leads to a reduction in polar night jet strength, which perturbs the polar vortex. Cold air outbreaks as well as warm air intrusions result in strong changes of surface temperature over days and the first month. Intrusion of warm, moist air masses into Arctic latitudes increases cloud cover, which reduces the amount of longwave radiation lost to space, leading to local surface temperature rise. Also the southern hemisphere shows a pattern of warming and cooling, however less pronounced, due to the positioning of the forcing.

After one year, Arctic amplification begins to emerge as an overall stronger cooling of the surface over Arctic latitudes compared to lower latitudes and the Antarctica. The ocean surface starts to cool down, although the decrease is still smaller compared to land surfaces.

The average over years 120–150 shows an overall cooling in all regions. Arctic Amplification is apparent as a stronger cooling over Arctic latitudes. North of Antarctica, where sea ice extent is increased compared to the piControl-runs, the surface

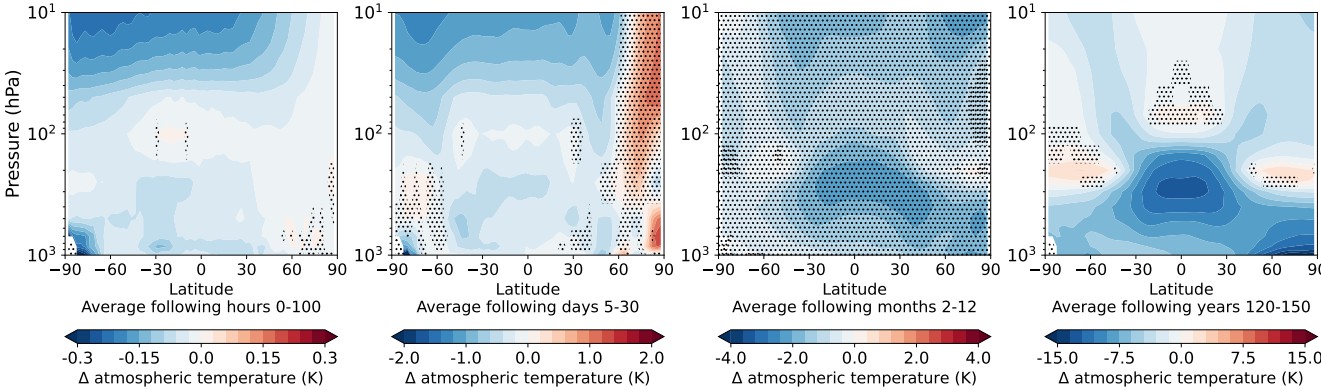

**Figure 5.** Multi-model mean, temporal and zonal mean vertical profiles of atmospheric temperature anomaly in K as a function of latitude for four different time scales after onset of forcing (100 hours, days 5–30, months 2–12, years 120–150). Regions, where less than three out of four models agree on the sign of anomaly, are dotted. Note the adjusted colour scales for the different time scales.

cooling is slightly stronger compared to the land surface of Antarctica, due to more severe changes in surface properties compared to the control run. The differences between land and sea surface are less apparent compared to the shorter time scales because also the deeper ocean adapts to the new energy balance.

Figure 5 shows the zonal mean of atmospheric temperature anomaly averaged over time scales of hours, days and months and the years 120–150, respectively.

During the first 100 hours, the stratosphere cools down quickly, especially above Antarctica, where the radiative forcing is strongest. Since the stratosphere is nearest to the source of forcing, the cooling of the stratosphere is particularly stronger than the cooling of the troposphere during the first hours and days. The cooling effects are strongest on the highest altitudes, where a lot of high frequency radiation is absorbed.

At the near surface layers, cooling above Antarctica is further amplified, because the surface temperature drops rapidly, which reduces sensible heat flux.

After one month atmospheric temperature change is highly variable in northern latitudes, due to the beforehand described perturbation of the polar vortex. Warm air intrusions into Arctic latitudes strongly increase the atmospheric temperature in surface near layers.

Over the first year, the troposphere adjusts to the solar forcing via an overall reduction of tropospheric temperature, especially in the mid and upper troposphere in the tropics. This pattern is similar to the long term pattern, where it produces a lapse rate feedback. However, similar atmospheric temperature adjustments were found for solar forcing experiments by e.g. Salvi et al. (2021), for fixed SST experiments. This indicates that this pattern also could be driven by changes in atmospheric absorption, humidity and convection in reaction to the forcing itself, rather than being a surface temperature mediated effect. We discuss this point further when analysing the results for humidity and cloud fraction.

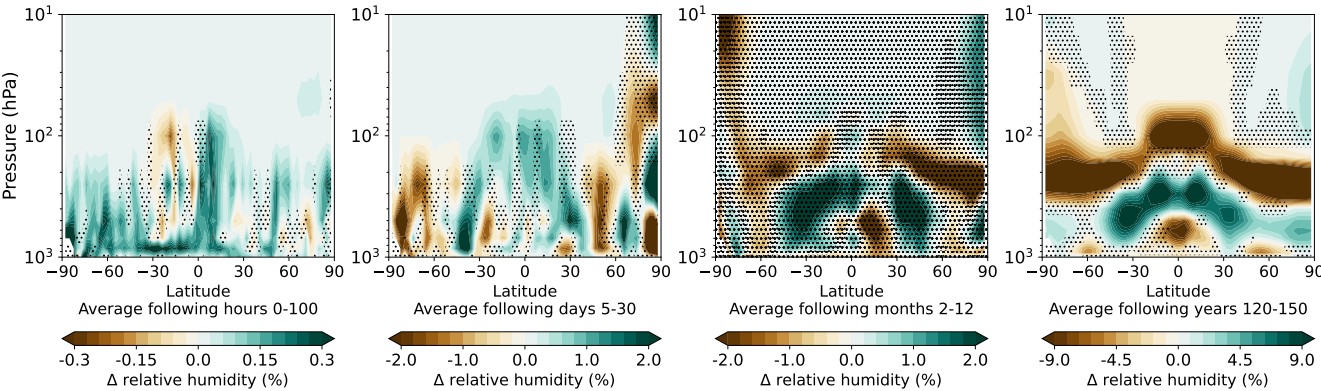

**Figure 6.** Same as Fig. 5 but for atmospheric relative humidity anomaly in %.

On longer time scales the tropospheric temperature continues to decrease. While in the tropics, this is most pronounced in the higher troposphere, at high latitudes the cooling is stronger in the lower troposphere, because the characteristic atmospheric temperature profiles in these regions react differently to a decrease in surface temperature.

In higher latitudes, the upper troposphere/lower stratosphere (UTLS) is shifted towards lower altitudes (visible when comparing absolute atmospheric temperatures of solm4p- and piControl-experiment). Hence, stratospheric vertical increase of temperature starts at lower altitudes, which manifests as a warming at $200\,\mathrm{hPa}$ at the poles. At $\pm 30°$ the stratosphere cools at all pressure levels.

### 3.2.2 Adjustments of humidity, vertical velocity and cloud fraction

Figure 6 shows the vertical distribution of zonal mean relative humidity anomaly. While specific humidity reacts to the cooling air temperature by decreasing in the whole atmosphere, especially strong in the tropics and over all time scales, relative humidity displays an overall increase during the first 100 hours, apart from a decrease in southern tropical latitudes. During the first month, an overall increasing pattern remains in the tropics, while higher latitudes show decrease. During the first year, the high troposphere dries, while the middle tropospheric layers experience a moistening. The lower troposphere shows a regular pattern of moistening and drying and only at $\pm 65°$ the whole troposphere dries. The pattern further intensifies on long-term time scales.

To further analyse the short-term adjustments of relative humidity, it makes sense to differentiate between trends on land and over sea, since they react very differently to the forcing on short time scales due to their different heat capacities. Hence, Fig. 7 shows the zonal means of relative humidity change for the first two time scales averaged over ocean (Fig. 7a) and land (Fig. 7b) only. To investigate the effects of changing vertical atmospherical stability Fig. 8 shows the same for vertical velocity change.

Over ocean, the lower troposphere cools down quicker than the sea surface on time scales of hours to days. This leads to a decrease in vertical stability, which then increases convective activity, especially over tropical oceans (see Fig. 8a). Increased

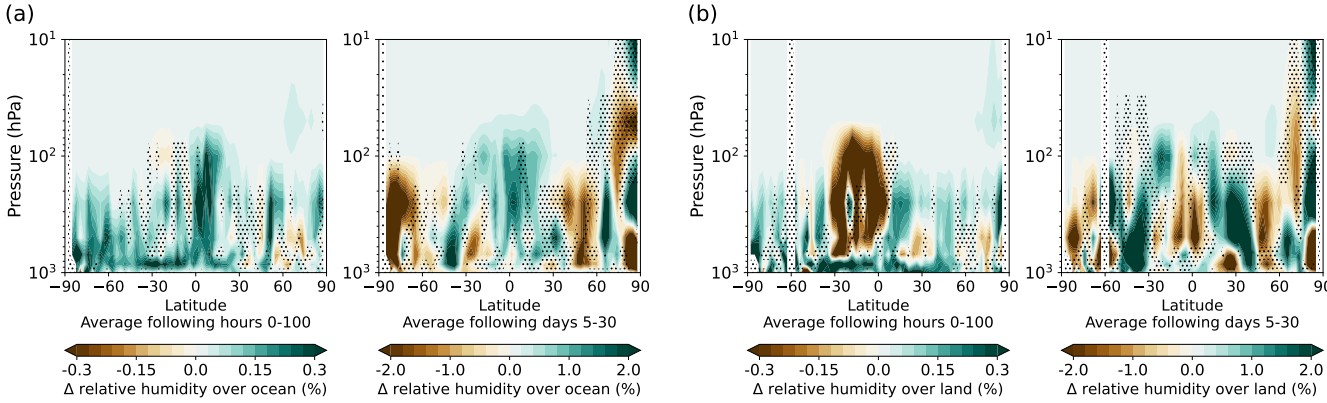

**Figure 7.** Zonal mean of relative humidity anomaly in % for the first 100 hours and days 5–30 (corresponds to the first two time scales of Fig. 5) averaged over ocean (a) and land (b) only.

convection dries the boundary layer, thereby reducing near surface specific humidity. In order to compensate for the deficit in humidity, evaporation and thereby latent heat flux increases which then results in an increase in surface relative humidity (see Fig. 7a) and precipitation in tropic latitudes. Cao et al. (2012) described very similar effects, although of opposite sign for $4xCO_2$ and $+4\%$ increase of solar constant experiments.

Over land, vertical stability does not show a clear sign and very much depends on location, albedo and heat capacity of the respective surface type. There is a strong reduction of convection over the tropics due to the reduced radiative forcing (see Fig. 8b) and resulting reduction in sensible heat flux, especially over the Sahara, which leads to trapping of moisture in the boundary layer and an increase in relative humidity in surface near layers. Latent heat flux is reduced in equatorial regions and overall less moisture is transported into the higher troposphere, which reduces relative humidity in the free troposphere (see Fig. 7b).

The significant differences between land and ocean surface response are strongest during the first hours after onset of forcing, but they are still clearly visible during the first month and lead to changes in land-sea-circulation.

For vertically-resolved atmospheric cloud fraction the four models only provided monthly data. Hence for the zonal mean of atmospheric cloud fraction anomaly, shown in Fig. 9) no data was available for the first time scale and the first monthly mean was plotted as the "first 31 days" time scale. This is not fully identical to other plots of this kind, because, if available, the daily data from day 5–30 was averaged, not including the first 4 days, while in case of the atmospheric cloud fraction, the average of the whole month is plotted.

As expected, the atmospheric cloud fraction anomaly pattern in Fig. 9 is overall very similar to the pattern of relative humidity in Fig. 6.

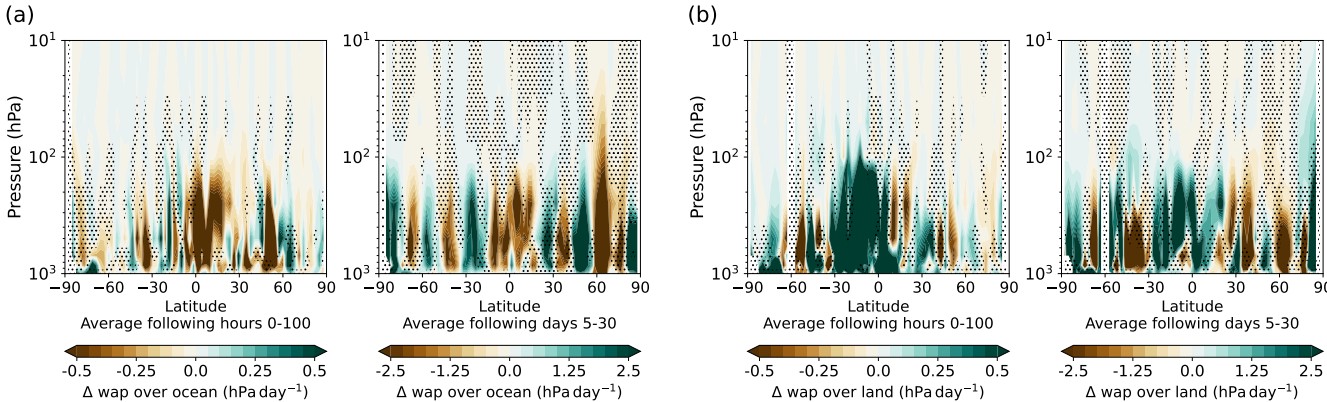

**Figure 8.** Same as Fig. 7 but for vertical velocity (wap) anomaly in $\mathrm{hPa\,day}^{-1}$.

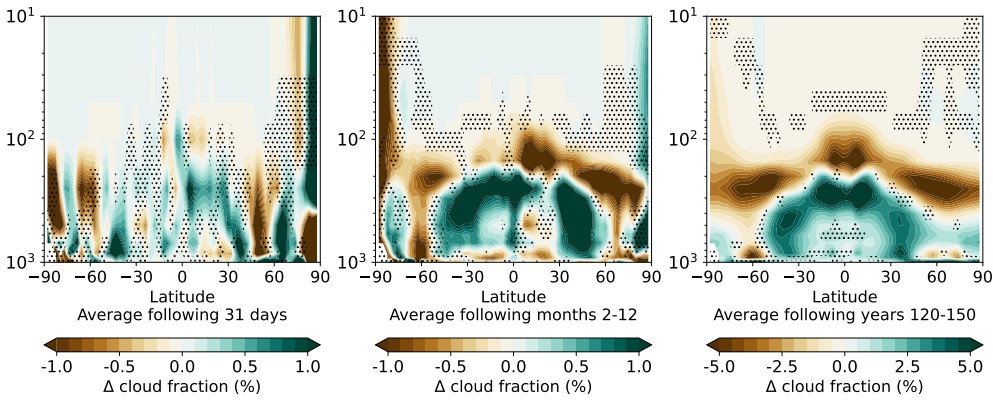

**Figure 9.** Same as Fig. 5 but for atmospheric cloud fraction anomaly in $\%$ on 49 pressure levels (the three other models were interpolated to CanESM5 levels).

During the first month, the atmospheric cloud fraction increases in the tropics over ocean, while it decreases over land.
However, the increase over ocean dominates the zonal mean. Higher latitudes show a pattern of decreasing and increasing cloud fraction, where circulation strength changes (also visible in the vertical velocity anomaly in Fig. 8).

Over the first year, tropical regions experience the strongest negative radiative forcing leading to less absorption of shortwave radiation in the troposphere and reduced convective heating from the lower troposphere. Because the colder air contains less water vapour, also the greenhouse effect is reduced, a positive feedback. This way, condensation levels are reached at lower
altitude and high clouds shift downwards. This leads to a decrease in cloud cover at $150\,\mathrm{hPa}$ and an increase in cloud cover around $250\,\mathrm{hPa}$, mirroring the same effects in relative humidity. An exception of the overall increasing cloud fraction in the troposphere is just above $\pm 60°$, where the subtropical jet stream strength is reduced, due to reduced temperature contrast

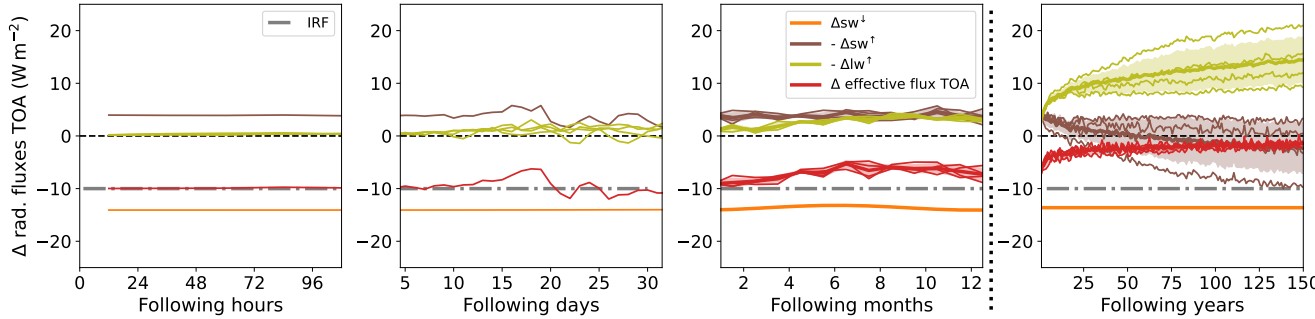

**Figure 10.** Global mean anomaly of radiative fluxes at TOA in $\mathrm{W\,m^{-2}}$ for four different time scales after onset of forcing (100 hours, 30 days, 12 month, 150 years). Plotted are the downward shortwave flux anomaly at TOA (orange), the upward shortwave flux anomaly at TOA (brown), the upward longwave anomaly at TOA (olive) and the effective flux anomaly at TOA (red). If available, the results for all four models (thin lines) were plotted together with the model-mean (thick line). In addition to that, the instantaneous radiative forcing (IRF) is plotted as a grey dash-dotted line. Upward fluxes were multiplied by $\cdot(-1)$, such that the three radiative flux anomalies (orange, olive and brown) add up to the effective flux anomaly at TOA (red), which is the same as the black curve in Fig. 3.

between mid- and high latitudes and weaker storm systems produce less dynamic lifting, reducing cloud formation in those regions.

On longer time scales, the overall pattern further intensifies, as do atmospheric temperature and relative humidity pattern.

     A distinct drying pattern in the relative humidity at the lower troposphere in the tropics (see Fig. 6), is not visible in the atmospheric cloud fraction, because it appears at a height of generally low cloud cover.

### 3.2.3    Effects on the radiative budget anomaly at TOA

Figure 10 shows the three components of the TOA radiative flux anomaly, downward and upward shortwave flux anomaly

(orange and brown) and upward longwave flux anomaly (olive), together with the effective flux anomaly at TOA (red, same as black curve in Fig. 3). Fluxes are defined positive downwards, so that added up, the three flux anomalies yield the effective flux anomaly. Only MRI-ESM2-0 provided TOA shortwave radiative fluxes as daily data, while the other three models only provided monthly data. Hence, Fig. 10 only shows the results of one model for shortwave flux anomalies and the effective flux anomaly for the first two time scales. The upward longwave radiative flux was provided as daily output by all models and is

plotted accordingly.

     The downward shortwave flux anomaly shows the reduction of the solar constant. It stays constant over all time scales and only displays a yearly periodic deviation due to the elliptical orbit of the Earth around the sun. It is the same for all four models, since the underlying assumption of the abrupt-solm4p experiment is an instantaneous and constant reduction of the solar constant by $4\,\%$, corresponding to a decrease of downward shortwave flux by $-13\,\mathrm{W\,m^{-2}}$.

The upward shortwave anomaly also shows an instantaneous reaction, because the planetary albedo initially stays the same, but the amount of incoming shortwave radiation that could be reflected is reduced. This instantaneous effect is part of the

instantaneous radiative forcing (IRF), because it is a change in TOA radiation in reaction to the reduction of the solar constant, without any changes in atmospheric state. Hence, the IRF is the sum of incoming and outgoing shortwave anomaly at $t = 0$ and is around $-10\,\mathrm{W\,m^{-2}}$ according to experimental design. The upward shortwave anomaly shows first changes after 10 days,

although no clear trend can be recognised. This is a reaction to changes in surface albedo (beginning after roughly 10 days and mostly connected to increased snowfall over land in high northern latitude and first changes in snow cover and sea ice extent in Antarctica) and cloud properties (beginning after a few hours, as described above). Although there are clear changes in albedo and cloud liquid water path and also variation in cloud cover over the first year, the upward shortwave anomaly stays relatively constant, indicating, that the different climate variables and local variability cancel out each other's shortwave radiative effects.

Over longer time scales, the multi-model mean of upward shortwave flux anomaly decreases, changes its sign around 50 years after onset of forcing and continues to decrease up until 150 years. However, the individual model runs do not agree on the sign of the anomaly on these longer time scales, due to opposing effects in high and low latitudes, discussed further below.

In contrast to the shortwave fluxes, there are no immediate effects of the reduction of the solar constant on the upward longwave flux. This is expected, because it is linked to changes in surface and atmospheric temperatures, which need some

400 time to adapt. Similar to the upward shortwave flux anomaly, the upward longwave flux anomaly displays first changes around 10 days after the onset of forcing. It is overall slightly increased over the following days and only temporarily changes its sign. The multi-model mean always stays positive, indicating a decreased loss of longwave radiation to space, because atmosphere and surface reduce their temperatures in reaction to the reduced incoming solar energy. This effect continues and all models simulate a further increasing longwave flux anomaly for longer time scales. The effective flux anomaly at TOA is overall

negative due to the negative forcing and corresponds to the IRF at $t = 0$. It is dominated by the shortwave effects on the time scales of hours and days, but in the course of month, the increasing longwave gain slowly reduces the TOA imbalance until it approaches a new balance over the following decades and centuries.

Figures 11a, 11b and 11c show the respective geographical distributions of the TOA radiative fluxes. Because only MRI-ESM2-0 provided daily data, the first time scales does not display a multi-model mean, but only the data of the MRI-ESM2-0

model. For the following 30 days, the January model mean of all four models were averaged.

Figure 11a clearly shows the uneven distribution of downward shortwave flux anomaly due to the chosen starting point of the experiment on 1 January. As the region of maximum sun exposure moves further north during the following month, the averages over the first year and the following 120–150 years then show the expected symmetric distribution after a full seasonal cycle with the strongest forcing in the tropics.

The upward shortwave flux in Fig. 11b reflects the distribution of the initial forcing, but locally adapted according to surface or cloud albedo. Especially Antarctica and the region of the westerlies with usually high cloud cover show the strongest reduction in upward shortwave flux. Similar to several other aforementioned climate variables, the uncertainty is high for the following 30 days. Only the reduction of upward shortwave flux in Antarctica remains relatively constant during the first month, due to the unchanged snow albedo. In contrast to that, South America shows an increased reduction in upward shortwave flux

compared to the surrounding regions and the first time scale. Since surface albedo remains relatively constant over the first month, this effect can be attributed to changes in cloud properties in reaction to an overall drying and reduced ascent over

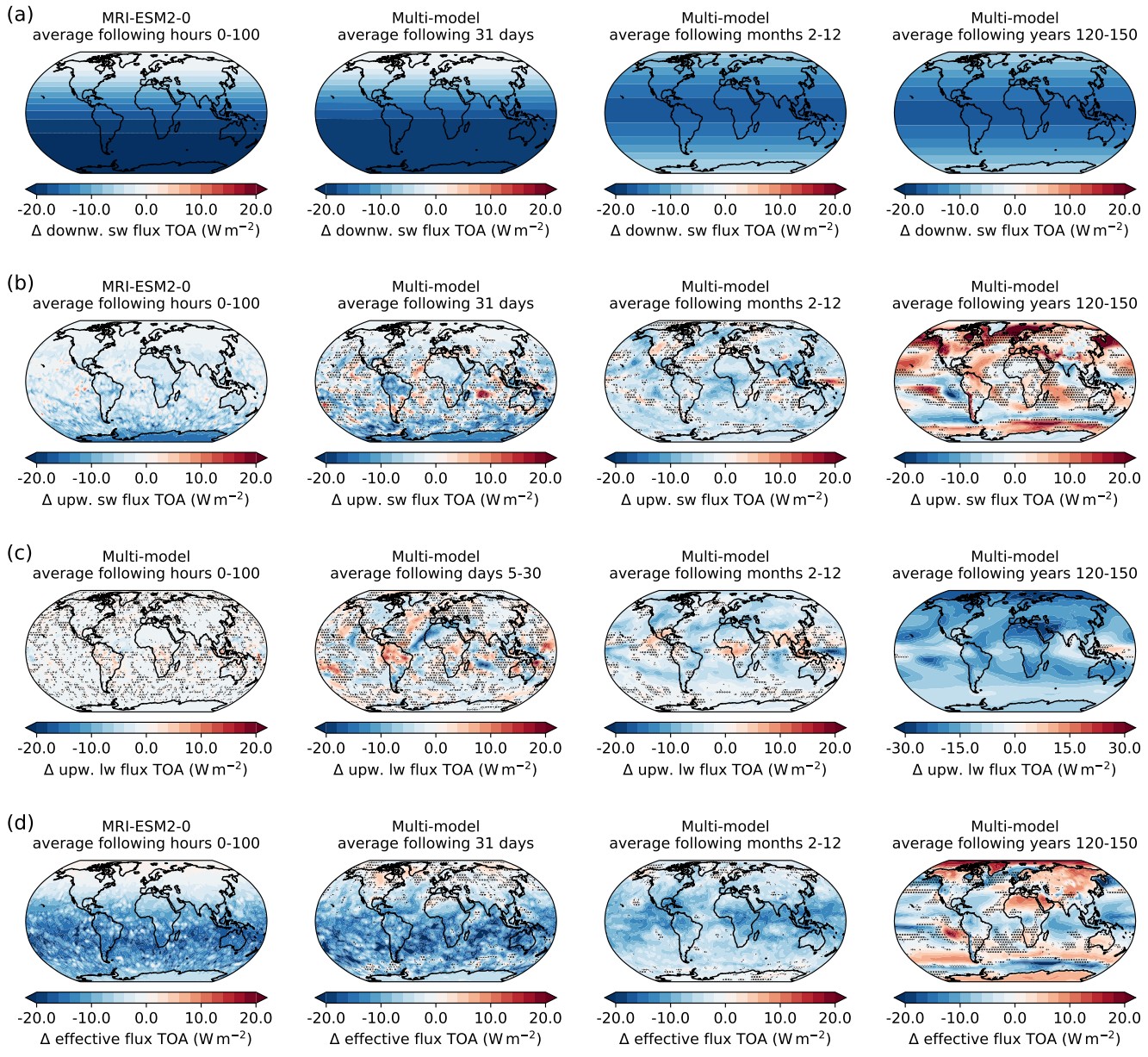

**Figure 11.** Multi-model mean global distribution of TOA radiative fluxes in $\mathrm{W\,m^{-2}}$ averaged over four different time scales after onset of forcing (100 hours, 31 days, months 2–12, years 120–150). Plotted are (a) the downward shortwave radiative flux anomaly at TOA, (b) the upward shortwave radiative flux anomaly at TOA, (c) the upward longwave radiative flux anomaly at TOA and (d) the effective radiative flux anomaly at TOA. For the first time scale of (a), (b) and (d) only data from MRI-ESM2-0 is plotted, because other models only provided monthly data. Regions, where less than three out of four models agree on a sign of anomaly are dotted.

tropical land areas. Hence, cloud liquid and ice water paths are decreased over South America, coinciding with an overall decrease in specific humidity and total cloud fraction. This leads to less scattering of downward shortwave flux and hence, more absorption by the surface. Because of its comparably low albedo (mostly green vegetation), the upward shortwave flux is reduced and hence, the planetary albedo is decreased. The opposite effect can be seen in the Indian Ocean, where liquid and ice water paths as well as total cloud fraction are increased, thereby increasing planetary albedo. During the first year, the overall negative trend of upward shortwave flux continues, with only local positive anomalies, where surface albedo or cloud properties are changed. On longer time scales, the cooling of the surface leads to sea ice spreading further to lower latitudes, which strongly increases surface albedo in the respective areas, leading to positive anomalies in upward shortwave flux. Similar surface albedo effects are visible on land, where snow cover is increased, especially in mountain areas. In contrast to that, increased upward shortwave flux over ocean is linked to an increase in cloud fraction, liquid and ice water path. Depending on whether the increase due to increased surface albedo or cloud cover, or the overall decrease due to the reduction of incoming shortwave radiation is stronger, the sign of the global mean upward shortwave flux anomalies differs between different models.

Figure 11c shows the anomaly of upward longwave radiative flux. Overall decreasing atmospheric and surface temperature lead to a decrease of outgoing longwave radiation with time. However, this effect is partly compensated by the reduction in water vapour. This process is especially effective in the tropics, where colder temperatures lead to a reduced amount of water vapour. This reduces the water vapour greenhouse effect and thereby optical depth of the atmosphere, which then in turn can result in a more effective radiative cooling. This can lead to slightly positive signals in outgoing longwave radiation over tropical land areas. Cloud property changes further amplify these effects. During the first two time steps anomalies are small and the uncertainty is high. The few stronger signals coincide with the beforehand described upward shortwave signals due to cloud property changes. The longwave effects often partially counteract the shortwave effects, because an increase in cloud liquid and ice water path will lead to more reflection of shortwave radiation, but at the same time, more upward longwave flux is absorbed, thereby decreasing TOA upward longwave flux. Over the first year upward longwave radiation further decreases due to the reduction of surface temperature, the so-called Planck effect. It becomes the dominant effect on longer time scales, clearly showing the effect of Arctic Amplification as a stronger cooling in Arctic latitudes. The only exception is an increase in upward longwave radiation northeast of Oceania, where cloud liquid and ice water path are reduced significantly, counteracting surface cooling effects.

Figure 11d displays the global distribution of the effective flux anomaly at TOA. During the first 100 hours, the reduced downward shortwave flux is the dominating influence on the TOA effective flux and the distribution of the shortwave forcing is clearly visible. Short-term adjustments reduce the overall loss of energy at TOA only locally, e.g. where precipitation is increased in the tropics, which coincides with changes in cloud properties. During the first month, the pattern remains relatively unchanged. Only the strong changes in surface temperature in high northern latitudes, due to the aforementioned disruption of the polar vortex, lead to a reduction in upward longwave radiation lost to space. Since the region is not directly influenced by the reduced solar constant, this longwave effect leads to a slight increase of the effective flux at TOA in high northern latitudes.

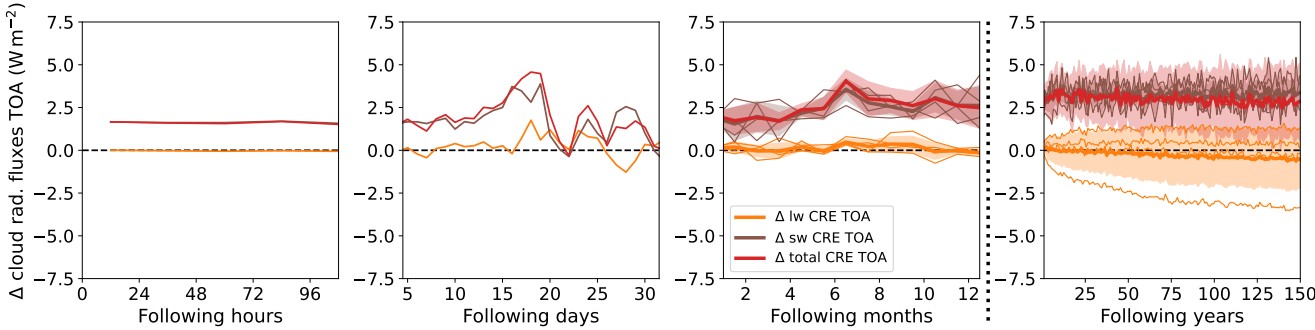

**Figure 12.** Global mean of TOA cloud radiative effect anomaly ($\Delta\mathrm{CRE}_{\mathrm{TOA}}$, red) in $\mathrm{W\,m^{-2}}$ for four different time scales after onset of forcing (100 hours, 30 days, 12 month, 150 years). Additionally, shortwave and longwave TOA cloud radiative effect anomalies ($\Delta\mathrm{CRE}_{\mathrm{TOA,sw}}$ and $\Delta\mathrm{CRE}_{\mathrm{TOA,sw}}$) are shown in brown and orange. Their respective intermodel standard deviation is shown as shading around inter-model means.

Over the first year, the TOA effective flux anomaly is still dominated by the reduced solar constant. Local effects due to changes in cloud properties do not have significant effects on the effective flux at TOA, because shortwave and longwave effects compensate each other.

On longer time scales, the TOA effective flux anomaly reaches a new balance and the global mean anomaly approaches zero. However, the global distribution of TOA effective flux anomaly shows a significant pattern over different latitudes, depending on the strength of local short- and longwave effects. The Planck effect generally acts against the radiative shortwave forcing by reducing the upward longwave radiation. Hence, the TOA effective flux anomaly is positive, where either the reduction in upward longwave radiation is stronger than the forcing or where also the upward shortwave flux anomaly is negative, counteracting the initial forcing.

### 3.2.4 Effects on the cloud radiative effect at TOA

Figure 12 shows the temporal evolution of the global mean cloud radiative effect anomaly at TOA ($\Delta\mathrm{CRE}_{\mathrm{TOA}}$) and its short- and longwave component ($\Delta\mathrm{CRE}_{\mathrm{TOA,sw}}$ and $\Delta\mathrm{CRE}_{\mathrm{TOA,lw}}$). As described before, only MRI-ESM2-0 provided daily data, which is plotted for the first two time scales.

The global multi-model mean (red) is positive for all time scales, indicating a decrease of the generally cooling effect of clouds, partially counteracting the negative forcing. However, since this positive anomaly is apparent from the first time step, the initial value is not attributable to changes in cloud properties. It rather is a result of clouds masking changes in the clear-sky budget. After five days $\Delta\mathrm{CRE}_{\mathrm{TOA}}$ starts to deviate from its initial value and varies with no clear pattern during the first month. During this time period it follows the same shape as the total TOA budget anomaly, hence, changes in cloud properties seem to be the dominant source of variability of the effective flux anomaly at TOA during the first months. This is supported by the fact, that global mean clear sky fluxes show near to no variability during the first five months. However, the peak of $\Delta\mathrm{CRE}_{\mathrm{TOA}}$ around six month coincides with a slight change in downward shortwave flux due to the Earth's elliptical orbit,

which also results in a drop in clear-sky upward shortwave flux. Therefore, this is more likely to be a masking effect, rather than purely attributable to changes in clouds, although some small changes are also recognisable in liquid and cloud water path. Moreover, after six month, the initial forcing moves from the southern high latitudes to the northern high latitudes. Due to differing surface albedo and therefore changes in the clear-sky budget, cloud masking is expected to change at this point.

After about 10 years $\Delta \mathrm{CRE_{TOA}}$ remains relatively constant at $4\,\mathrm{W\,m^{-2}}$. A slight decreasing trend is visible in the multi-model mean, even though three out of for models simulate a constant $\Delta \mathrm{CRE_{TOA}}$ for longer time scales. This is due to the continuous decrease of $\Delta \mathrm{CRE_{TOA,lw}}$, simulated by the CanESM5 model, which is not visible in any other model.

If cloud properties were to stay constant, the pure cloud masking effect would lead to a negative $\Delta \mathrm{CRE_{TOA,lw}}$, because clouds would dampen the Planck effect, which is the main process counteracting the negative forcing. This dampening effect would be stronger, the stronger the surface temperature reduction was. Changes in cloud properties can either increase the dampening (e.g. due to an increase of liquid or ice water path) or counteract the dampening, if absorption of upward longwave radiation by clouds is reduced. Hence, the sign of $\Delta \mathrm{CRE_{TOA,lw}}$ depends on whether changes in cloud properties can compensate the cloud masking effect of the Planck effect or not. Only in case of the CanESM5 model, the masking effect is much stronger than the effects of cloud property changes. Hence, it simulates a negative long-term trend for $\Delta \mathrm{CRE_{TOA,lw}}$.

Figure 13 shows the global distribution of the total $\Delta \mathrm{CRE_{TOA}}$ (Fig. 13c), as well as its shortwave and longwave components, $\Delta \mathrm{CRE_{TOA,sw}}$ and $\Delta \mathrm{CRE_{TOA,lw}}$ (Fig. 13a and 13b).

During the first 100 hours, $\Delta \mathrm{CRE_{TOA,sw}}$ is strongest on the southern hemisphere, where the shortwave forcing is strong and cloud cover is high. Hence, the increased $\Delta \mathrm{CRE_{TOA,sw}}$ is mostly an effect of cloud masking, rather than caused by changes in cloud properties. The only exception is Antarctica, where the high snow albedo suppresses any possible shortwave cloud radiative effects. In contrast to that, $\Delta \mathrm{CRE_{TOA,lw}}$ remains relatively unchanged and only in the tropics some stronger signals appear, roughly coinciding with areas of increased precipitation, indicating increased condensation and bigger droplets that absorb more longwave radiation and lead to a positive $\Delta \mathrm{CRE_{TOA,lw}}$. Vice versa, areas of decreased total cloud fraction over central Africa, Peru and northern Australia lead to negative $\Delta \mathrm{CRE_{TOA,lw}}$.

During the first month, $\Delta \mathrm{CRE_{TOA,sw}}$ further strengthens and the beforehand mentioned decrease in cloud liquid and ice water path over South America as well as the opposite effect on the Indian ocean are visible as areas of increased and decreased $\Delta \mathrm{CRE_{TOA,sw}}$, respectively. The same regions show up in the $\Delta \mathrm{CRE_{TOA,lw}}$ as regions of stronger signal, but of opposite sign due to the aforementioned compensation of short- and longwave effects. Moreover, a region of increased cloud ice water path and cloud fraction on the west coast of Africa produces a clear signal of decreasing $\Delta \mathrm{CRE_{TOA,sw}}$ and increasing $\Delta \mathrm{CRE_{TOA,lw}}$, since less shortwave radiation is scattered and more longwave radiation is transmitted to space.

Over the first year $\Delta \mathrm{CRE_{TOA,sw}}$ remains overall positive, with a more homogeneous distribution over the globe, due to the more homogeneous distribution of the forcing after a complete seasonal cycle. Only Oceania shows a distinct decrease of $\Delta \mathrm{CRE_{TOA,sw}}$ and increase of $\Delta \mathrm{CRE_{TOA,lw}}$, where all cloud properties, analysed in this study, show a distinct reduction. However, as described before, short- and longwave effects compensate each other in a way, that the changes in cloud properties, mentioned above, are not visible in the total $\Delta \mathrm{CRE_{TOA}}$.

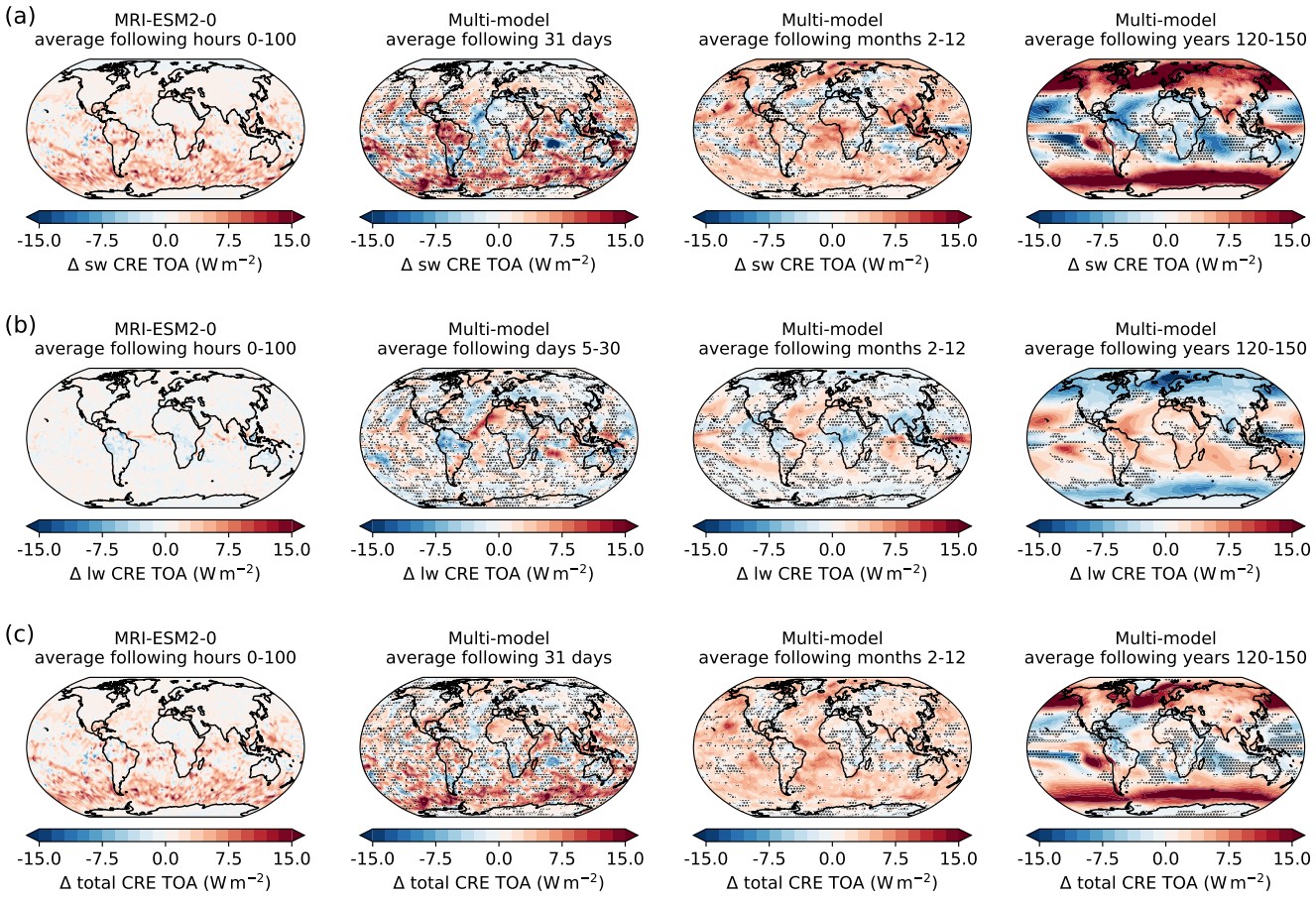

**Figure 13.** Same as Fig. 11 but for (a) shortwave and (b) longwave and (c) total TOA cloud radiative effect anomaly ($\Delta\mathrm{CRE}_{\mathrm{TOA,sw}}$, $\Delta\mathrm{CRE}_{\mathrm{TOA,lw}}$ and $\Delta\mathrm{CRE}_{\mathrm{TOA}}$) in $\mathrm{W\,m^{-2}}$.

On long-term time scales, $\Delta\mathrm{CRE}_{\mathrm{TOA,sw}}$ shows a clear correlation with increased surface albedo at high latitudes, leading to a strong positive masking effect. Moreover, decreased cloud liquid water path, e.g. over high latitudes and the equatorial Pacific lead to increase of $\Delta\mathrm{CRE}_{\mathrm{TOA,sw}}$ in these regions, while an increasing cloud liquid water path over tropical oceans has the opposite effect. In contrast to that, $\Delta\mathrm{CRE}_{\mathrm{TOA,lw}}$ shows stronger correlation with the cloud ice water path anomaly, since ice clouds have a generally stronger longwave effect than liquid clouds. In total, shortwave effects are overall stronger and thus, dominate $\Delta\mathrm{CRE}_{\mathrm{TOA}}$, while longwave effects only appear at certain locations of strong longwave anomalies. Hence, a separate consideration of short- and longwave effects can help to understand the underlying processes better, compared to only considering the total $\Delta\mathrm{CRE}_{\mathrm{TOA}}$. Nevertheless, masking effects impede a clear detection of cloud adjustments based on the cloud radiative effect.

## 4 Discussion

The aim of this study is to understand and analyse atmospheric processes that take place on time scales of hours, days and months after the onset of a solar forcing. We did so using the output of four different climate models that participated in the abrupt-solm4p experiment of CFMIP of CMIP6. These fully coupled simulations provide the opportunity to get a more realistic few on rapid adjustments in the climate system, as all components of the climate system are allowed to react to the forcing at the same time, rather than fixing (sea) surface temperatures. This allows for a more realistic interplay between the different components, although a $+4\%$ solar forcing still needs to be considered a highly idealized forcing compared to e.g. transient climate change.

Our approach is based on the assumption, that the short timescales of hours, days and months are dominated by rapid adjustments rather than by surface temperature mediated changes. We showed in Fig. 3 that for these three time scales, the simulated change in surface temperature cannot explain the simulated change in TOA fluxes. Only when adjusting it by the rapid adjustments found in the linear regression plots of Sect. 3.1, temperature mediate TOA flux change and simulated TOA flux change start to coincide after around 4 years in model mean. This corresponds to the time, the linear regression plots show non-linearities and is likely connected to adjustments of system with high inertia, like cryosphere and vegetation or a delay in response of the deeper ocean compared to the surface. Since our study concentrates on adjustments during the first year, these kind of adjustment are not considered here.

Having atmosphere, surface and deep ocean react to the forcing at the same time, poses the challenge of a possible overlap of adjustments to the forcing and temperature mediated processes. Nevertheless, our findings do agree in many points with findings of other studies that applied a variety of different methods, often more restrictive compared to our approach, which we discuss in the following.

Smith et al. (2018) analysed adjustments to five different forcing agents, among them a $+2\%$ solar forcing, using 11 GCMs with fixed SST in combination with radiative kernels. Similar to our results, they found that in case of solar forcing, rapid adjustments overall reduce ERF. However, the adjustments we found using the linear regression method were overall stronger in relation to the original forcing than they were in case of Smith et al. (2018). According to their findings, the change in surface, tropospheric and stratospheric temperature adjustment contribute the most to counteracting the forcing. Our method does not allow a quantification of the different effects in the same way, but we also see a decrease in temperature for surface as well as atmospheric temperatures in global mean, which will reduces the TOA forcing. The overall atmospheric cooling leads to a reduction of specific humidity in the whole atmosphere, which reduces the warming properties of water vapour in the atmosphere and is hence expected to increase TOA forcing, which is also shown in Smith et al. (2018). We find a very similar pattern of cloud fraction change to that reported by Smith et al. (2018), though of opposite sign, since this study was based on a reduction of the solar constant. We calculated the anomaly of cloud radiative effect in order to estimate the influence of cloud adjustments on the ERF. However, the cloud radiative effect anomaly contains masking effects which hampers the comparability with studies, that specifically quantified cloud adjustments. Nevertheless, when assuming, that all initial signals

in $\Delta CRE_{TOA}$ can be attributed to masking, rather than changes in cloud properties, this initial signal can be subtracted, leaving a more variable, but positive cloud adjustment, like it was found by Smith et al. (2018).

Russotto and Ackerman (2018) conducted a study using G1 data from the geoengineering MIP (GeoMIP) of CMIP6, where a fourfold $CO_2$ increase is balanced by a matching decrease in solar constant. The decrease is between $-3.2\%$ to $-5\%$, hence of similar magnitude as was used in the abrupt-solm4p experiment. That experimental setup avoids a long term change in surface temperature. Hence, all signals are a mix of adjustments to the two different forcing agents. They find a significant cooling of the higher troposphere, as was shown in our results and a warming of the polar regions, which we also detected.

However, in their case it is probably linked to the so-called "residual polar amplification", which can be seen in only $CO_2$ forcing experiments and is of higher magnitude and higher SNR than the signal, found in this study. A significant difference in the atmospheric temperature change is the stratospheric cooling, which is much stronger in the results of Russotto and Ackerman (2018) due to adjustments to 4x$CO_2$ because of the increased LW radiation to space (Manabe and Wetherald, 1975). This effect is much stronger than the stratospheric cooling in response to solar forcing, which we see in our results. Although of

similar magnitude, cloud fraction changes in the G1 experiment seem to be dominated by adjustments to $CO_2$, since Russotto and Ackerman (2018) find opposite signals to what we found. This is supported by the findings of Smith et al. (2018), who estimate cloud adjustments to $CO_2$ forcing to be twice as large as to a positive solar forcing with twice as strong IRF. However, Smith et al. (2018) quantified a positive solar forcing and Aerenson et al., 2024 show, that responses to positive and negative solar forcing can differ, not only in sign, but also in strength and pattern.

Furthermore, for their solp4p experiment Aerenson et al. (2024) find an increasing cloud cover over tropical land regions, decreasing cloud cover in higher latitudes over land and an overall reduction in cloud cover over ocean. We find similar patterns, though of opposite sign, as expected for a reduction of solar constant. The same is true for the CRE components they calculated for a solp2p experiment. However, when applying their newly developed method, Aerenson et al. (2024) found an overall positive cloud radiative adjustment to a positive solar radiative forcing of $+4\%$ of solar constant, meaning that clouds

further increase the initial solar forcing rather than reducing it, mostly via their shortwave effects due to changes in cloud fraction. This is the opposite effect compared to what was found by e.g. Smith et al. (2018) or Virgin and Fletcher (2022) and what our results indicate. Aerenson et al. (2024) show that there can be substantial differences in adjustments to positive and negative solar forcing and linearity cannot be assumed. Nevertheless, they also find the opposite sign to Smith et al. (2018), who also analysed positive solar forcing simulations. All studies found high uncertainty for cloud adjustments and, similar

to our study, Aerenson et al. (2024) only had data from four climate models they could base their analysis on. In their case one model predicts negative cloud adjustments and only two models show significantly positive adjustments. Moreover, their method combines fixed SST with fully coupled model runs in order to quantify adjustments to solar forcing. Several studies have shown, that the two approaches, although showing similar results, can also show deviations due to differences in e.g. dynamical adjustments of land-sea-circulation. The approach of Aerenson et al. (2024) may mistakenly interpret $CO_2$ forcing

offsets caused by the fixed SST method as adjustments to solar forcing.

     All in all this disagreement shows how challenging a proper quantification of cloud adjustments is and that they remain a source of high uncertainty in climate models.

Salvi et al. (2021) conducted a study on adjustments, using idealized atmospheric heating experiments, which were then fitted to the diagnosed heating curve of different forcing agents. This way, they found that the vertical center of mass, what they call the "characteristic altitude" determines the cloud adjustments. Like Smith et al. (2018) and our study they found negative cloud adjustments to in their case a $+3\%$ solar forcing. In their observed and fitted data they found a decrease in relative humidity and cloud fraction at around 300 and $800\,\mathrm{hPa}$, which corresponds well with the increased cloud fraction we find in these altitudes. There is some deviation in the UTLS region, where we find a significant reduction in cloud fraction and relative humidity, while Salvi et al. (2021) only diagnosed minor changes in cloud fraction and the fitted algorithm predicted a decrease. This again could either be an effect of different methods, as Salvi et al. (2021) themselves attribute the deviation to insufficient resolution of the vertically heating curves in these high altitudes. Nevertheless, some differences in response to positive and negative solar forcing can take place, like Aerenson et al. (2024) found, especially when considering non-linear processes like phase changes in clouds.

Moreover, our study supports the findings of other studies like Kamae et al. (2019), that adjustments can also depend on the season of initialization. The strong response in Arctic surface temperatures, which we attribute to a disruption of the polar vortex is an effect of initializing the forcing in January.

A challenge our study faced was the sparse dataset it is based on. Only four models participated in the solm4p experiment, each providing one run. A higher number of models and ensembles would increase the reliability of our findings. However, due to the strong nature of the forcing and since every model initialized the experiment from a different point in their piControl runs, we do expect, that the four models cover different climate variability modes. Hence, an agreement of several models on the sign of an adjustment effect and overall high model-mean anomalies indicate a signal that exceeds possible climate variability. For all analysed climate variables, the first month is the time scale with lowest agreement of the models on the sign of the signal. This is to be expected, since responses will strongly depend on the position of e.g. pressure systems at the moment of onset of forcing. For example, this becomes apparent in the Arctic surface temperature anomaly during the first month, were all models simulated strong changes in surface temperature. We attribute those to a disruption of the polar vortex, but the actual pattern differs between the models, since it will be influenced by the original state of atmosphere. Nevertheless, Fig. 4 shows strong signals for several high latitude regions, were at least 3 out of 4 models agree on the sign. Therefore, we interpret it as a rapid adjustment rather than a random signal due to climate variability.

One surprising aspect of this study was the disagreement of the long term development of the models, especially when examining cloud related variables. While the models agree well on the TOA budget anomaly, they differ in how the long- and shortwave components contribute to this and how the fluxes interact with changes in cloud properties. In a number of cloud variables and hence, in the cloud radiative effect anomalies, high latitudes and tropics show an opposite response and the global mean signal is determined by the dominating effect, which differs between the models. Further investigation into the single models and parametrization as well as possible tuning, would be necessary to understand the different outputs. Though this was out of the scope of this study, it shows, how even climate models of the newest generation still struggle to simulate short and long term change of cloud properties and further research on the topic is necessary.

## 5 Conclusion

This study offers insights into the dynamics of short-term adjustments within the climate system in response to an instantaneous radiative forcing. This was done through the analysis of the abrupt-solm4p experiment of CMIP6. By simulating a $4\%$ reduction in the solar constant, the experiment provides a valuable framework for understanding how the Earth climate system reacts to changes in solar energy input, especially on shorter time scales of days to month.

The analysis reveals significant alterations in various climate variables, including surface and atmospheric temperatures, relative humidity and vertical velocity as well as a number of cloud properties. All models simulate decreasing atmospheric and surface temperatures, beginning immediately after the onset of forcing in January. Significant differences were found between the response of relative humidity over land and over ocean. Over ocean, decreased vertical stability, due to a quicker cooling of the lower troposphere compared to the ocean surface, leads to increased convection and thereby drying of the boundary layer, which is then compensated by an increased latent heat flux leading to an increase in relative humidity. Over land in the tropics, convection is reduced and moisture effectively trapped in the boundary layer, which leads to an increase of relative humidity in surface-near layers and a decrease in relative humidity in the higher troposphere. This highlights the complex interactions and differences between land and sea response. On time scales of months, overall lower temperatures lead to drying of the troposphere and the decrease in solar energy in the tropics leads to a reduction of ascend, less convection and a descending high cloud layer.

Inter-model agreement of cloud variables was overall low. Generally, clouds are a major source of uncertainty in climate models due to complex interactions of a huge number of variables and non-linear processes. This was especially obvious, when analysing the total cloud radiative effect anomaly. While the single short- and longwave components showed characteristic correlation with a number of cloud variables like cloud liquid and ice water path, the total cloud radiative effect anomaly is dominated by cloud masking effects. Characteristic regions of changing cloud properties over South America and the Indian ocean were less discernible in the total cloud radiative effect anomaly, because short- and longwave effects partially compensated each other. Therefore, a separate analysis of short- and longwave effects can support process understanding. A higher number of models participating in the experiment would be beneficial, in order to further reduce uncertainty. Nevertheless, significant signals were found for e.g. for atmospherical cloud fraction and our findings where in good agreement with other studies on the topic.

The TOA rapid adjustments was quantified via linear regression and RA of $3.6\,\mathrm{W\,m^{-2}}$ were found, reducing the initial radiative forcing of $10\,\mathrm{W\,m^{-2}}$ by $36\%$. Other studies like Smith et al. (2018) also found that RA were counteracting an initial solar forcing. However, the amount of reduction differs, based on the method and number of participating models.

While other experiments of CMIP6 are designed in a way, that influence of seasonal and climate variability are reduced by using a number of different initialization dates, the abrupt-solm4p simulations use only one date (1 January). This could be viewed as a limitation of this study. Nevertheless, if the scientific community strives for a validation of new discoveries about rapid adjustments in climate models with e.g. satellite measurements, more realistic simulations are crucial in order to further improve state of the art climate models. This study showed that the positioning of the forcing can have significant influence on

the resulting short-term adjustments, in this case, the disruption of the polar vortex and consequently warming of the higher northern latitudes.

Moreover, the described adjustments to a reduced solar constant may resemble adjustments to more realistic forcing scenarios. After large volcanic eruptions a global stratospheric aerosol scattering layer can form a few months after the eruption. This aerosol layer reduces the amount of shortwave radiation that reaches the surface, which could lead to similar adjustment effects as simulated for a reduced solar constant. However, in case of the abrupt-solm4p experiment, the forcing is instantaneous and constant, which makes it easier to differentiate between instantaneous forcing, short-term adjustments and long-term feedbacks. Therefore, it is a useful tool on the way to a more in depth understanding of realistic forcing agents of more transient nature. By elucidating the short-term adjustments that occur in response to reduced solar radiation, essential insights can be provided that can inform risk assessments associated with solar radiation management methods. Understanding these adjustments is vital for anticipating potential local impacts, such as droughts or floods, which can have significant consequences for communities and ecosystems.

In conclusion, this study identifies several characteristic adjustment processes as well as a number of challenges related to the current approaches in quantifying adjustments. By integrating short-term adjustment processes into climate modelling efforts, the scientific community can improve the accuracy of long-term climate predictions and develop more effective strategies for addressing the challenges posed by climate change.

*Data availability.* The CFMIP model data from the abrupt-solm4p experiment used in this study are freely available from the CMIP6 repository on the Earth System Grid Federation nodes (https://esgf-metagrid.cloud.dkrz.de/search/cmip6-dkrz/, World Climate Research Programme, 2020).

*Author contributions.* CL and JQ designed the study. CL analysed the CFMIP abrupt-solm4p experiment data, produced the figures, and drafted the initial manuscript. Both authors contributed to the writing, editing and review of the paper.

*Competing interests.* JQ is a member of the editorial board of Atmospheric Chemistry and Physics

*Acknowledgements.* This research has been supported by the Deutsche Forschungsgemeinschaft Research Unit VolImpact (FOR2820, QU 311/23-2) within the project VolCloud. This work used resources of the Deutsches Klimarechenzentrum (DKRZ) granted by its Scientific Steering Committee (WLA) under project ID bb1036.

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
