# Peer review of "Adjustments to an abrupt solar forcing in the CMIP6 abrupt-solm4p experiment"

_EGUsphere, 2024_

## Author Response (AR1)

**Authors' response**

**Review #1**

We thank the reviewer for their time and the feedback to our manuscript. We attempted to address all comments and answer them in detail below.

This paper examines simulations of the abrupt-solm4p model experiment from four Earth system models that contributed to CMIP6. Specifically, the authors investigate the models' respond over different timescales to an abrupt reduction of the solar constant. Through the majority of the paper they focus on the climate response on sub-daily, daily, monthly, and centennial timescales. Toward the end of the paper the authors use a linear regression with global mean surface temperature to calculate the effective radiative forcing, and instantaneous radiative adjustment in the top-of-atmosphere radiative fluxes and the cloud radiative effect output of the abrupt-solm4p model simulations. Although many of the results presented in this paper are novel and interesting, there are some deficiencies in this paper that make it not currently suitable for publication. Below I have listed my major comments that need to be addressed before this paper should be published.

We thank the reviewer for their very good summary of our manuscript.

1. The structure of the manuscript is flawed, where the introduction bears little relevance to most of the results shown. The title and introduction are about rapid adjustments and effective radiative forcing, while until Section 3.6 and figure 14 there are no results shown on the rapid adjustments or effective radiative forcing. Instead, nearly the entire paper focuses on the models' response at different timescales and does nothing to distinguish the rapid adjustments from temperature mediated changes. Due to this discrepancy between the introduction and nearly the entire results section, there is not a cohesive story to the paper. This makes it difficult to understand why the results shown are significant and how they relate to the rest of the paper. I recommend the authors re-organize and rewrite a lot of the material to either be about the different timescales of response to solm4p, or about the rapid adjustments and ERF.

We thank the reviewer for their comment. In light of this comment, we have substantially restructured the manuscript for a revised version and made sure the flow of the descriptions are clearer now.
Differentiating between rapid adjustments and feedbacks in simulations without fixed SST is challenging and we agree with the reviewer that more attention should be given to this point. We addressed this in our revised manuscript and restructured the paper in the following way.

(1) We added a paragraph to the Introduction, which not only mentions previous studies on solar forcing, but also summarizes their main findings *(Lines 80-103 in the new version of the manuscript)*

(2) We shifted the quantification of rapid adjustments via the linear regression method to the beginning of the results section *(now Sect. 3.1)*. This way, the results section starts more

consistently with the approaches of studies discussed in the Introduction, namely the regression analysis and thus classical way to quantify rapid adjustments in fully coupled climate models.

(3) We also added a new section *(Sect. 3.2)* that addresses the reviewer's concern about the link between adjustments and analysis of response over different time scales. There, we show that multiplying the observed temperature change with the climate sensitivity derived from the linear regression cannot explain the change of TOA effective flux simulated by the models during the first three time scales. Only when applying an initial offset (i.e., the rapid adjustments), the long term development is explained this way. These added results demonstrate that the first three time scales are dominated by rapid adjustment processes.

(4) We then continue with the results part as structured before *(Sect. 3.2.1 to 3.2.4)*, analysing the adjustments of different climate variables over these first three time scales. Here we apply a broader definition of rapid adjustments and consider all alterations to the climate system in response to the forcing that are independent of global surface temperature change as adjustments, following the literature.

(5) We removed parts of the former *Section 3.3* "Effects on cloud properties". As the paragraphs on total column integrated cloud liquid and ice water path were mostly of descriptive nature, we decided to omit them in the revised version of the manuscript for the sake of better flow of reading and to avoid unnecessary length of the manuscript.

(6) We added an explicit discussion section *(Sect. 4, more details in answer two second comment)*

2. This paper is missing substantial literature review. This would include summarizing some of the most relevant studies in the introduction and explaining in the conclusions how the present results relate to the previous literature. I believe that this manuscript would benefit from discussion of how the changes in abrupt-solm4p relate to changes from CO2 forcing, and other studies that have looked at solar forcing. I would like to point the authors to a pair of recently published studies on models' response to solar forcing that they might find relevant.

Aerenson, T., & Marchand, R. (2024). Cloud Responses to Abrupt Solar and CO2 Forcing: 1. Temperature Mediated Cloud Feedbacks. Journal of Geophysical Research: Atmospheres, 129(12), e2023JD040296. https://doi.org/10.1029/2023JD040296

Aerenson, T., Marchand, R., & Zhou, C. (2024). Cloud Responses to Abrupt Solar and CO2 Forcing: 2. Adjustment to Forcing in Coupled Models. Journal of Geophysical Research: Atmospheres, 129(12), e2023JD040297. https://doi.org/10.1029/2023JD040297

We thank the reviewer for the comment and for providing specific references. We added the adjustment paper to the introduction *(Line 79)*. Furthermore, we added a paragraph in the introduction, which describes the findings of previous studies on solar forcing in more detail than was done in the previous version of the manuscript *(Lines 80-103)*.

The submitted manuscript did not include an explicit discussion section. Based on further literature review and the papers, including the ones kindly provided by the reviewer but also

including several others, we now added an extra discussion section *(Lines 520-621)* that more explicitly discusses the findings in the context of other studies' findings *(Lines 540-601)*. Moreover, limitations of this study's approach are now discussed in more detail in the new Discussion section *(Lines 602-621)*.

3. The manuscript is also lacking explanation of how the authors distinguish the forced response from internal variability. On timescales as short as hourly, daily, and yearly one would imagine that the modeled climate response would be susceptible to internal variability. As far as I can tell the manuscript does not include any description of how the authors remove the variability signal to isolate the forced response. Without doing so, one cannot determine if the response is due to changing phases of for example, ENSO or the NAO instead of a response to the change in the solar forcing. If the authors are doing something to distinguish the forced response from internal variability, it needs to be mentioned in the manuscript. Or alternatively, if the authors can show that the internal variability is much smaller than the forced response that would also be adequate but would also have to be shown in the manuscript.

We do agree with the reviewer, that short time scales are especially susceptible to internal variability of the models. Unfortunately, the output of only four models that participated in the solm4p experiment of CMIP6, each providing one run, is available. This is indeed quite sparse to base a detailed analysis of internal variability on. However, due to the strong forcing of 10 W m-2, the forced changes in the atmosphere still may exceed natural variability, as AR6 quantified anthropogenic forcing in 2019 to be of the order of 2.7 W m-2 and Sippel et al., 2021 showed, that this already allows a robust detection of forced warming, significantly exceeding climate variability.
An attempt was made to account for internal variability in our study by only considering areas in the discussion, where three out of four models agreed on the sign of the signal. We do acknowledge that this can not replace a thorough significance analysis, which unfortunately was not possible based on the sparse data available for this study. We agree that this shortcoming was not acknowledged enough in the manuscript. We addressed this in the new discussion section *(Lines 602-613)*.

4. The manuscript mentions "not shown" results a handful of times, nearly all of which had me wondering why it is not shown. Generally, if a result is important enough to discuss it is important enough to show. I understand that this paper is already quite long, so some of this could be provided in a supplement.

We thank the reviewer for pointing this out. As the reviewer supposed, we decided to not show some results, because of the length of the paper, if the results were very much expectable and did not further contribute to the discussion of new findings. Nevertheless, we wanted to mention these results for the sake of completeness. However, since the reviewer kindly made us aware that this might cause more confusion than insight, we decided to take the respective passages out of the revised manuscript. Since the revised manuscript contains two additional parts (TOA budget anomaly from global mean surface temperature change and the new discussion section), which makes the paper even longer, we also decided to leave out the sections on

cloud liquid and ice water path *(former Lines 263-299)*, since they were mostly descriptive and did not contribute significantly to better understanding adjustment processes.

**Review #2**

We thank the reviewer for their time and their comments on our manuscript. We want to use the opportunity to address the comments in detail below.

This paper analyzes the adjustment of the climate system to a 4% reduction in incoming solar radiation in CMIP-6 abrupt-solm4p simulations. Although the paper does contain some interesting results, a few points require clarification and some further investigations before the study can be accepted for publication.

We thank the reviewer for their concise summary of our study.

Main comments:

1) Internal variability: As far as I understand, the simulations analyzed amount to 8 runs: 4 control runs and 4 experiments. Certainly, there are global phenomena active on that timescale (ENSO) with amplitudes of a few tenths of K, which should partly hide the signature of the adjustment. Could the authors discuss it in more detail?

We thank the reviewer for pointing this out. Indeed, addressing internal variability is an important point when discussing rapid adjustments and proves especially challenging in simulations that do not use fixed sea-surface temperatures. However, especially those can provide new inside in rapid adjustments in more realistic scenarios and are hence of interest to the scientific community. As the reviewer pointed out, this study was based on the solm4p experiment by CMIP6, for which only four different models provided output data, each with only one run. This is, as the reviewer rightly points out, a very sparse base to address internal variability. However, due to the strong forcing of 10 W/m-2 we expect forced alterations of the atmosphere to exceed natural variability considerably, as AR6 quantified anthropogenic forcing in 2019 to be of the order of 2.7 W/m-2 and Sippel et al., 2021 showed, that this already allows a robust detection of forced warming.

An attempt was made in the manuscript to account for internal variability, by only considering areas where three out of four models simulated the same sign of signal. Nevertheless, we acknowledge that this cannot be considered a full significance analysis and means that the uncertainty of the results needs to be considered. We added a new discussion section to the revised manuscript, which addresses this shortcoming more clearly than it was done in the previous manuscript *(Lines 602-613)*.

2) Model uncertainty: There is a significant intermodel spread in the longer-term climate response (in temperature, humidity, etc.). However, the pattern of the spread, as well as

possible reasons, are not discussed.

We agree with the reviewer, that especially cloud related variables showed strong inconsistencies in their long term behaviour between the four participating models. However, since this study concentrated on the short term adjustments, which overall showed more consistency between the models. We assume, that long term differences are a among others a result of different cloud parametrizations, tuning and resolution between the models. One effect that we found, which strongly influenced the global mean response of cloud variables, was a reemerging contrast between tropics and high latitudes. Depending on which effect dominated in the respective model, the global mean could show different signs. Unfortunately, addressing these differences was out of the scope of this study and we hope for future research to shed more light on this issue. Nevertheless, we also addressed this disparity in the new discussion section *(Lines 614-621)*.

Minor comments and typos:
Figure 1, caption of panel 3: modelmean -> model mean; xlabel: month -> monthS

Thank you for pointing this out, we changed the caption accordingly.

line 495: A parallel is drawn between the rapid adjustment to a reduction of the solar constant and natural perturbations of the stratospheric aerosol layer, induced by, e.g., a volcanic eruption. However, the timescale of dispersion in the stratosphere is several months, such that the rapid adjustments are unlikely to be relevant. This should be considered in the text.

We thank the reviewer for this comment. The overall aim of this work is to learn more about rapid adjustments. Since in most realistic situations, the forcing is neither instantaneous nor constant, examining rapid adjustments is a big challenge, as the reviewer rightly points out. This we plan to address by starting with very simplified experiments, like the reduced solar constant and then gradually moving onto more realistic simulations. As pointed out by the reviewer, volcanic forcing is changing over the course of months, when the stratospheric aerosol is distributed over the globe and after a few years goes back to zero, when the aerosol deposited and/or was washed out.
We now specified in the respective text passage, that in order to build a link from this idealised simulation to a realistic one considering volcanic aerosol, one has to consider the different time scale of the evolution of the forcing and with it of the adjustments *(Lines 658-662)*. We added a new section to the revised manuscript, in which we show that temperature mediated TOA budget change cannot explain the simulated change in TOA budget for the first three time scales of hours, days and months *(Sect. 3.2)*. In a similar way, it might be possible to show that also in case of volcanic eruptions the first months are dominated by adjustment processes, rather than temperature mediated processes. Since, also significant global mean surface temperature change requires the stratospheric aerosol layer to cover a considerable amount of the Earth, we still expect to see a time delay between adjustment and feedback processes. In

this case, we would not so much be interested in aerosol cloud interactions that happen quickly after a volcanic eruption, but rather in larger scale adjustments of circulation, cloud variables and surface temperature patterns, as we found in this study. We plan to address this comparison in a future study and hope that it helps to further elucidate how the Earth's climate system adjusts to more realistic forcing scenarios.

---

## Referee Report (RR1)

Review of *Adjustments to an abrupt solar forcing in the CMIP6 abrupt-solm4p experiment*, Lange and Quaas

This paper looks in detail at the evolution of the climate system in response to an abrupt reduction in 4% incoming shortwave radiation. The authors have addressed several of the key points from previous reviews, but some minor details should be adjusted before publication.

**Main Comments:**

Figure 1: it appears by eye that the yearly anomalies plotted follow a gradient in color. Based on the plots I can only assume that the gradient corresponds to the year plotted, with darker years being older? This should be clarified in both the caption and potentially a colorbar added to the figure.

Lines 225-235: The authors discuss nonlinearities in the response of SW up, a departure from the linear behavior of the first decade. However, these nonlinearities do not jump out from the referenced Figure (1c). Differentiating the first 10 years of the response in the figure could remedy this.

Lines 239-246: The mechanisms behind the counteracting cloud adjustments should be discussed. Are these dominated by changes in BL clouds? Do these adjustments come from reductions in the optical thickness of such clouds, or changes in the areal coverage of clouds, and do the 4 models agree on the origin of these adjustments? As mentioned, perhaps the magnitude differs among the ensemble, but the different contributions to the cloud adjustments may be partitioned similarly across models.

Sect 3.2.1: The response in the first month of the forcing is interesting, particularly the warming over the Arctic, where it is claimed that warm, moist air intrusions drive increased cloud cover and a reduced of LW radiation emitted to space, warming the local surface temperature. However, in Figs 4b and 5b, it appears that this region is subject to an uncertain response, or that not all models simulate the warming. The authors should discuss why some of the models don't simulate this warming, because presumably all 4 models simulate a reduced meridional temperature gradient, which is put forth as the initial driver of perturbations to the polar vortex. It seems like this gets a very brief mention in line 610, but I think this should be more prominent in the text and not buried in the discussion.

Figure 8: The authors should specify the sign convention for vertical velocity (wap).

Figure 10: The different behavior of the models over time in terms of both LW and SW up is interesting, especially when comparing the timescales at which individual models seem to stabilize, whereas one model exhibits a continually decreasing SW up and increasing LW up throughout almost the entire duration of the simulation. Do the authors have insight as to this model's differing behavior?

Line 344: How do the estimates of surface RH changes in the tropics from Cao et al (2012) compare in terms of magnitude to those in this study?

Line 650: Please contextualize the Smith (2018) results by providing the percentage by which RA reduce the initial radiative forcing, if possible.

The comments from previous reviewers noting the non-negligible role of internal variability remain applicable. The discussion section has helped clarify some of the shortcomings of the relatively small ensemble used here. The justifications in the review response by the authors should perhaps be included in the main text.

**Typos:**

Line 28: "number of studies on the subject *have been* conducted"

Line 70: While it can be inferred that the abrupt-solm4p experiment is a simpler analogue of volcanic eruptions, the transitions between these two paragraphs should make this more explicit and clear

Line 81: "to decreased temperature of *the* troposphere…"

Lines 95-100: In reporting the results of other literature, the authors state that SW and LW effects tend to cancel each other out, with SW effects slightly stronger. The next sentence then states that SW effects dominate the overall cloud adjustments. Can the authors make clear how this is the case, when it is stated that the SW and LW tend to counteract each other and cancel?

Line 108: change development to *evolution*

Line 117: "that all rapid adjustments happen *while the global mean.."* (no comma needed)

Line 119: "while global mean surface temperature *has already begun to change*"

Line 120: "On the 120 other hand, the fixed surface temperature method (Hansen et al., 2005) or rather fixed-sea-surface temperature (SST), which is easier to implement in global climate models (Forster et al., 2016), is widely used and has the advantage of suppressing feedbacks." Suggest moving Forster citation to the end, and remove clause about ease of implementation. It is somewhat implied in the wide use of this method that it is simple to implement.

Line 125: "or cooling cannot be simulated in *this kind of setup"*

Line 134: The meaning of the sentence leading with "Thermodynamical and dynamical" is unclear, suggest rewording.

Lin 208: "slope than in *the* long term" "developing *for* as long as a decade"

Line 218: "budget, the yearly mean"

Line 402: "indicating a reduction in the amount of longwave radiation lost to space"

Line 531: "temperature mediate*d*"

Line 533: I'm not sure what the authors are trying to convey with the sentence leading with "This corresponds to the time", suggest rewording to emphasize the role of high inertia components of the climate system

---

## Author Response (AR2)

**Authors' response**

**Review #1**

We thank the reviewer for their time and their feedback to our revised manuscript. We attempted to address all comments and answer them in detail below.

This paper looks in detail at the evolution of the climate system in response to an abrupt reduction in 4% incoming shortwave radiation. The authors have addressed several of the key points from previous reviews, but some minor details should be adjusted before publication.

We are happy to have addressed the intended key points of the first review and tried to also address the new points in the following.

Main Comments:

Figure 1: it appears by eye that the yearly anomalies plotted follow a gradient in color. Based on the plots I can only assume that the gradient corresponds to the year plotted, with darker years being older? This should be clarified in both the caption and potentially a colorbar added to the figure.

We thank the reviewer for pointing this out and added a sentence to the caption of Fig. 1, explaining how the color gradient shows the temporal evolution of surface temperature and TOA budget change.

Lines 225-235: The authors discuss nonlinearities in the response of SW up, a departure from the linear behavior of the first decade. However, these nonlinearities do not jump out from the referenced Figure (1c). Differentiating the first 10 years of the response in the figure could remedy this.

We thank the reviewer for this comment. We shortened the manuscript according to the 2nd reviewer's comments and we decided to not show the linear regressions of the single fluxes in the main part of the paper anymore. Hence, also the discussion of the non-linearites is more brief than before. We added the SST-pattern effect as a possible cause for the observed non-linearity, but as the non-linearities appear on time scales of years while the paper concentrated on adjustments over the first days and months, this effect was not discussed in further detail.

Lines 239-246: The mechanisms behind the counteracting cloud adjustments should be discussed. Are these dominated by changes in BL clouds? Do these adjustments come from reductions in the optical thickness of such clouds, or changes in the areal coverage of clouds, and do the 4 models agree on the origin of these adjustments? As mentioned, perhaps the

magnitude differs among the ensemble, but the different contributions to the cloud adjustments may be partitioned similarly across models.

We thank the reviewer for this comment. Attributing contributions of different cloud property changes to the overall cloud adjustment is definitely an interesting endeavour and can improve understanding. However, this was not possible with the methods used for this paper as we can only discuss changes in the cloud properties and hypothesize on their interactions, but we cannot quantify their contributions to the overall cloud adjustments. Nevertheless, we added a sentence discussing how the decrease in boundary layer clouds at 850hPa coincides with strong positive signals of CRE_sw and hence, might be one of the causes of the overall positive cloud adjustments *(Lines 230-233)*.

Sect 3.2.1: The response in the first month of the forcing is interesting, particularly the warming over the Arctic, where it is claimed that warm, moist air intrusions drive increased cloud cover and a reduced of LW radiation emitted to space, warming the local surface temperature. However, in Figs 4b and 5b, it appears that this region is subject to an uncertain response, or that not all models simulate the warming. The authors should discuss why some of the models don't simulate this warming, because presumably all 4 models simulate a reduced meridional temperature gradient, which is put forth as the initial driver of perturbations to the polar vortex. It seems like this gets a very brief mention in line 610, but I think this should be more prominent in the text and not buried in the discussion.

We thank the reviewer for pointing this out. Indeed all models simulate a warming of Arctic regions (single models not shown in the paper). However, the specific location of warm air intrusions strongly depends on the base state of the climate, which is different for all four models. Hence, even though all models simulate an increase in Arctic surface temperature, they do not agree on the location, and thus, in some Arctic regions less than 3 out of 4 models agree on the sign. We added a sentence on this point in the newly revised manuscript *(Lines 370-373)*.

Figure 8: The authors should specify the sign convention for vertical velocity (wap).

We thank the reviewer for making this point, as the sign convention for the vertical velocity can be confusing, especially when looking at anomalies. We added a sentence that clarifies the sign convention and also decided to change the colorbar to a diverging colormap between blue and red. This leads to more intuitive visualizations as blue colors are associated with an increase in upward velocity or a decrease of downward motion. Moreover, we decided to combine Figures 7 and 8 from the previous version (relative humidity and vertical velocity averaged over land and ocean) as the discussion of the underlying mechanisms switches back and forth between the two variables.

Figure 10: The different behavior of the models over time in terms of both LW and SW up is interesting, especially when comparing the timescales at which individual models seem to stabilize, whereas one model exhibits a continually decreasing SW up and increasing LW up

throughout almost the entire duration of the simulation. Do the authors have insight as to this model's differing behavior?

We agree with the reviewer that the different behaviour of models for the long term time scale is surprising and raises new questions. However, for this paper we concentrated on the short term time scales and thus, decided to not go into detail on this long term deviation. Therefore, apart from the differing climate sensitivities of the four participating models, we unfortunately cannot give a more specific explanation as to what are the reasons for the long term behaviour. The only thing that we can point out is that the model with the strongest surface temperature response (CanESM5) is also the model that simulates the strong decrease in CRE_lw change. Therefore, this is more likely to be cloud masking effect, rather than overall stronger changes in cloud properties, as the model did not systematically show the strongest changes in other cloud properties. We added this discussion to the revised manuscript *(Lines 336-339)*.

Line 344: How do the estimates of surface RH changes in the tropics from Cao et al (2012) compare in terms of magnitude to those in this study?

We thank the reviewer for asking for clarification on this point and added the magnitude of Cao et al., 2012 findings to our revised manuscript and also elaborated more on the differences between their findings and ours *(Lines 434-438)*.

Line 650: Please contextualize the Smith (2018) results by providing the percentage by which RA reduce the initial radiative forcing, if possible. The comments from previous reviewers noting the non-negligible role of internal variability remain applicable. The discussion section has helped clarify some of the shortcomings of the relatively small ensemble used here. The justifications in the review response by the authors should perhaps be included in the main text.

We thank the reviewer for their comment and added specific numbers on Smith et al., 2018 findings to the conclusion *(Line 571)*. We also included the justification that we provided to the reviewers in the first revision round to the paper and thank the reviewer for this idea *(Lines 342-348)*.

Typos:

We thank the reviewer very much for carefully studying our revised manuscript and pointing out typos, which we corrected accordingly.

Line 28: "number of studies on the subject have been conducted"

We corrected this in *Line 26*.

Line 70: While it can be inferred that the abrupt-solm4p experiment is a simpler analogue of volcanic eruptions, the transitions between these two paragraphs should make this more explicit and clear

We added a sentence in *Lines 61-62* to clear this up.

Line 81: "to decreased temperature of the troposphere…"

We corrected this in *Line 75*.

Lines 95-100: In reporting the results of other literature, the authors state that SW and LW effects tend to cancel each other out, with SW effects slightly stronger. The next sentence then states that SW effects dominate the overall cloud adjustments. Can the authors make clear how this is the case, when it is stated that the SW and LW tend to counteract each other and cancel?

We thank the reviewer for their comment. The wording was indeed unclear in the manuscript and we adapted the sentence in order to avoid confusion *(Line 95)*.

Line 108: change development to evolution

We adapted the sentence accordingly in *Line 111*.

Line 117: "that all rapid adjustments happen while the global mean.." (no comma needed)

We deleted the unnecessary comma in the sentence *(Line 119)*.

Line 119: "while global mean surface temperature has already begun to change"

We changed the sentence in *Line 121*.

Line 120: "On the 120 other hand, the fixed surface temperature method (Hansen et al., 2005) or rather fixed-sea-surface temperature (SST), which is easier to implement in global climate models (Forster et al., 2016), is widely used and has the advantage of suppressing feedbacks." Suggest moving Forster citation to the end, and remove clause about ease of implementation. It is somewhat implied in the wide use of this method that it is simple to implement.

We thank the reviewer for their comment and implemented it in the suggested form in *Lines 122-123*.

Line 125: "or cooling cannot be simulated in this kind of setup"

We corrected the typo in *Line 127*.

Line 134: The meaning of the sentence leading with "Thermodynamical and dynamical" is unclear, suggest rewording.

We thank the reviewer for pointing this out and revised the sentence. We also restructured the Introduction section to improve clarity and avoid redundancy. Hence the paragraph on types of adjustment signals was shifted to an earlier point in the Introduction *(Lines 78-80)*.

Lin 208: "slope than in the long term" "developing for as long as a decade"

We corrected the manuscript accordingly in *Line 206*.

Line 218: "budget, the yearly mean"

We revised the sentence in *Line 213*.

Line 402: "indicating a reduction in the amount of longwave radiation lost to space"

We changed the wording according to the reviewer's suggestions as it is clearer in meaning *(Line 271-272)*.

Line 531: "temperature mediated"

We corrected the typo in *Line 472*.

Line 533: I'm not sure what the authors are trying to convey with the sentence leading with "This corresponds to the time", suggest rewording to emphasize the role of high inertia components of the climate system

We added more information to make the meaning of the sentence clearer *(Lines 472-476)*.

Review #2

We thank the reviewer for their time and their comments on our manuscript. We want to use the opportunity to address the comments in detail below.

This paper analyses the rapid adjustments to a 4% abrupt reduction of the solar constant across four global climate models, analysing the responses on timescales from hours to years. The analysis seems carefully done, and the focus on the time evolution of the adjustments is novel and useful. This paper can therefore be a useful addition to the literature, although I would request major revisions to address the shortcomings described below.

We thank the reviewer for their summary of our manuscript and address the requested revisions in the following.

Main comments:

My main criticism is that the presentation of the results can be improved: the paper is rather lengthy, lacks focus on the key results and could do a much better job highlighting the novel aspects of the findings. I would encourage the authors to try and identify 2-3 "key points", and make sure that these are highlighted in the abstract, results and conclusions, with a consistent narrative.

We thank the reviewer for their comment. In light of this remark, we worked thoroughly on the presentation of the results. We now shift the paper's focus stronger to cloud adjustments and characteristic patterns of change, which we motivated via the variability of TOA budget change and CRE change. Further following this comment, we also restructured the manuscript by shifting the global mean temporal development plots of TOA budget change and CRE after the linear regression section. We hope that this responds to the reviewer's remark to provide key findings and allows for a clearer narrative on why adjustments of atmospheric temperature, humidity and clouds are crucial to better understand the underlying mechanisms and are of interest, even though this method does not allow a quantification of the single variables to the total rapid adjustments at TOA. We expect to find the identified fingerprints also in more realistic forcing scenarios like volcanic eruptions. In those cases, it can help to decide, whether a signal is an actual adjustment signal or only climate variability, as the patterns identified here, were found for four models, which each started from a different base climate.

The issue starts with the abstract, which is too long (I counted ~280 words, which exceeds the APC word count limit of 250) and contains some repetitive points about the significance of rapid adjustments. The first two paragraphs need to be shortened and the abstract should then quickly get to the key points. The subsequent summary of the results is also too vague – for example the final sentence reads "On longer time scales we find robust changes of cloudiness". Either this is an important result and the text should then specifically describe what these

changes are and their implications, or I would leave this out from the abstract. Similar comments apply to the conclusions section.

We thank the reviewer for pointing this out and shortened the abstract accordingly. We now pay closer attention to the cloud fraction changes that were found in the analysis.

Given the paper is about rapid adjustments, it's a bit surprising that no numbers are provided in the abstract at all. The conclusions provide some numbers, but only in the fourth paragraph and almost as a casual side point. I would suggest stating these numbers near the start, and briefly comparing with prior work.

We thank the reviewer for their comment and quantified the rapid adjustments found via the linear regression method in the abstract. Moreover, we restructured the conclusion, paying closer attention to these findings and providing rapid adjustments found by Smith et al., 2018 for comparison *(Line 571)*.

As mentioned previously, the paper feels rather long. In highlighting the key results, the authors could also choose to shorten or cut other parts of the results section, including some of the figures. My overall impression from a quick read of the manuscript was that a lot of results were described whose significance was not immediately clear.

We thank the reviewer for their remark. With the restructuring of the paper described above, we decided to omit the global distribution plots of TOA budget change and CRE. These plots did not add substantially to the understanding of the underlying mechanisms of cloud adjustments, which we chose to be the main focus of the revised paper. Nevertheless, some interesting signals were found, like the partial compensation of short- and longwave cloud effects. Hence, we decided to provide the plots in the Supplements.

Finally, I would recommend that the authors perform a quick spelling and grammar check, as there are quite a few minor (but slightly annoying) mistakes. Please also check punctuation, particularly where relative clauses are involved (e.g. "we show, that" → "we show that"; "in cases, where" → "in cases where", etc.).

We thank the reviewer for taking the time to carefully read our manuscript and pointing out spelling and grammar mistakes. This helped a lot to further improve the text. We addressed all following comments by adapting the text accordingly.

Other minor comments:

- Why only consider the m4p experiment? Perhaps motivate this briefly in the introduction.

The aim of future work is to compare the findings of the solm4p experiment to modelling of the pinatubo eruption (VolMIP, volc-pinatubo-full). Since the stratospheric volcanic aerosol layer scatters part of the incoming shortwave radiation, we expect similar rapid adjustments in both

cases, although the different nature of the forcing (non-instantaneous and located in the stratosphere, rather than at TOA) will surely lead to some differences as well. Analysing similarities as well as differences between the responses to the two forcings could further expand our understanding of the underlying mechanisms.

Moreover, as Aerenson et al., 2024 showed in their work, solm4p and solp4p do not lead to the same adjustments of opposite signs, but there are significant differences due to non-linearities in the adjustment processes. Hence, also using p4p might not increase statistical reliability, but rather add new questions, which were not the focus of this study.

We added a sentence addressing this topic in *Lines 102-106*.

- L210: "inertia of the ocean" – this is vague. My feeling is that there is probably an "SST pattern effect" involved, as this is known to cause changes in the slope of N versus T over time.

We thank the reviewer for pointing this out. We added the SST pattern effect as a possible explanation of the observed non-linear behaviour and reworded the sentence to avoid being too vague *(Lines 205-212)*.

- L210: Can't the IRF be calculated directly? It's a 4% solar constant reduction, so the IRF should be 0.04*(solar constant)*(planetary albedo).

Theoretically, it would be possible to calculate the IRF directly using the solar constant and the planetary albedo. Then the IRF would be

0.04*(0.25*solar constant)*(1-planetary albedo)

However, this requires knowledge of the current planetary albedo at the moment of forcing. Since three out of the four models only provide monthly data, calculating the planetary albedo directly for the 01/01/1850 is not possible in an exact way. Because assumptions would have to be made either way, we decided to approximate the IRF by the values of the first month after the onset of forcing. The advantage of this approach is its applicability to the single radiative flux components as well as to the CRE. We acknowledge, that this approach is not exact, but as the variability between the models is much higher than any uncertainty that would be introduced by e.g. taking the mean albedo between December and January of the piControl run and calculating the IRF via the equation given above, we decided to rather use the first month response as an approximation of IRF.

- L256: I don't understand this sentence. Why "more than 1K"? And I think you mean "changes in the TOA radiative imbalance", not "forcing"

Thank you for this comment. The wording of the sentence was indeed confusing and we revised the paragraph *(Lines 286-291)*.

- L270: That's plausible, but hard to tell from the figure because with just four realisations (one per model), the timeseries are probably dominated by internal variability. A more convincing

argument (in my view) would be that the lambda*T term (which is not shown, but can be inferred from the evolution of the red and blue curves) is comparatively small in the first three panels. As an aside, it's a bit surprising that internal variability isn't mentioned at all in the discussion of this and other results (at least not before the discussion section towards the end of the paper).

We thank the reviewer for their comment. Internal variability is one of the major challenges when investigating rapid adjustments, especially when looking for characteristic local patterns rather than at global means. When using climate models, this can be addressed by running ensembles rather than single runs. However, as the reviewer pointed out, in the case of the solm4p only four models participated, each contribution only one realisation. This makes for less statistical significance than would be desirable.
However, we argue that the initial forcing of 10 W m^-2, is strong enough to trigger adjustments that exceed internal variability. This is based on findings of AR6, which quantified anthropogenic forcing in 2019 to be of the order of 2.7 W m^-2 and Sippel et al., 2021 showed that this already allows for a robust detection of forced warming, significantly exceeding climate variability.
We decided to discuss this issue already in Section 3.2 when discussing different time scales of adjustments *(Lines 306-312 and 342-349)*.

- L315: This could also be an effect of land surface temperature change, since land surface temperatures are interactive in fixed-SST runs.

We thank the reviewer for pointing this out. Possible influence of land surface temperature changes were not discussed when mentioning the findings from Salvi et al., 2021. We added a sentence about this topic, discussing how land surface temperature can change in fixed SST-experiments. However, the effects are small due to the coupling of ocean and land surface temperature via the atmosphere. Hence, we argue that, although possibly influencing the results, land surface temperature probably are not the main cause of the signal found by Salvi et al., 2021 *(Lines 396-399)*.

- L365: I would explain this more simply as a lowering of the tropopause.

We thank the reviewer for their comment. Since the tropopause does not descend everywhere, originally we decided not to use this explanation, but if specified for the high latitudes, it is correct and a more intuitive explanation than the one given before. We changed the text accordingly *(Lines 405-407)*.

- L393: The text mentions cloud water path here and elsewhere, but I don't think it's shown in any of the figures – consider cutting, or add "not shown".

We deleted this paragraph as it does not fit the new structure of the revised manuscript.

- Table 2: It would be good to report the CRE adjustments (and their range) with the cloud masking removed.

We thank the reviewer for this idea and added another column to Table 2 with the respective contribution to adjustments.

- Fig. 3: The caption doesn't describe the red shading. The individual curves for the ERF case are all blue, instead of using the colours corresponding to each model.

By reorganising the Figures, we address this now in the caption of the Fig. 3, which shows the temporal development of TOA radiative fluxes.
"Shading around the multi-model mean shows multi-model standard deviation."

---

## Author Response (AR3)

**Authors' response**

**Remarks from the preceding review file validation**

You added a supplement but the sections and figures are not named correctly. During the next revision please rename the sections and figures as explained at: https://www.atmospheric-chemistry-and-physics.net/submission.html#assets > supplements. The references to the supplementary material in your manuscript also need to be updated.

After double checking with the guidelines on supplementary material, we decided to provide the additional plots and paragraphs as an Appendix A (Linear regression plots for all TOA flux changes and CRE changes) and an Appendix B (geographical distribution on different time scales of TOA flux anomalies and CRE anomalies, as well as its short- and longwave components). These additional figures and sections are now named using the Copernicus Latex template.

Dear Authors,

I find that you have satisfactorily addressed the referees' comments. It is therefore my pleasure to accept the paper for publication; congratulations!
I have selected 'technical corrections' in order to give you the possibility to proofread the whole manuscript before it is transferred to copy-editting.

Sincerely
Aurelien Podglajen

We very much thank the editor for his time and the work on our manuscript! We used the opportunity to correct minor typos. Moreover, we cleared up the sign convention in section 3.1 and Table 2, as the previous version might have led to confusion in this regard. Furthermore, we updated Fig. 5, so each model would be shown in their characteristic linestyle, as was done in the other figures of this kind, and also the single model results for the total CRE anomaly were added for the sake of consistency and completeness.